# DO LLM AGENTS HAVE REGRET?
# A CASE STUDY IN ONLINE LEARNING AND GAMES

**Chanwoo Park**[⋆1]**, Xiangyu Liu**[⋆2]**, Asuman Ozdaglar**[1]**, Kaiqing Zhang**[2]
[1] MIT, [2] University of Maryland, College Park

## ABSTRACT

Large language models (LLMs) have been increasingly employed for (interactive) decision-making, via the development of LLM-based autonomous agents. Despite their emerging successes, the performance of LLM agents in decision-making has not been fully investigated through quantitative metrics, especially in the multi-agent setting when they interact with each other, a typical scenario in real-world LLM-agent applications. To better understand the limits of LLM agents in these interactive environments, we propose to study their interactions in benchmark decision-making settings of online learning and game theory, through the performance metric of *regret*. We first empirically study the no-regret behaviors of LLMs in canonical non-stochastic online learning problems, as well as the emergence of equilibria when multiple of them interact through playing repeated games. We then provide some theoretical insights into the sublinear regret growth in the cases we observed, under certain assumptions on (supervised) pre-training and the data generation model. Notably, we also identify (simple) cases where advanced LLMs such as GPT-4 fail to be no-regret. To further promote the no-regret behaviors, we propose a novel *unsupervised* training loss, the *regret-loss*, which, in contrast to the supervised pre-training loss, does *not* require the labels of (optimal) actions. Finally, we establish the *statistical* guarantee of generalization bound for regret-loss minimization, and more importantly, the *optimization* guarantee that minimizing such a loss can lead to known no-regret learning algorithms, when single-layer self-attention models are used. Our further experiments demonstrate the effectiveness of our regret-loss, especially in addressing the above "regrettable" cases.

## 1 INTRODUCTION

Large language models (LLMs) have recently exhibited remarkable emerging capabilities (Bubeck et al., 2023; Achiam et al., 2023; Wei et al., 2022b; Yao et al., 2023a). As a consequence, a burgeoning body of work has been investigating the employment of LLMs as central controllers for (interactive) decision-making, through the construction of *LLM-based autonomous agents* (Hao et al., 2023; Shen et al., 2023; Yao et al., 2023b; Shinn et al., 2023; Wang et al., 2023d; Significant Gravitas, 2023). Specifically, the LLM agent interacts with the (physical) world in a *dynamic/sequential* way: it uses LLMs as an oracle for reasoning and planning, then acts in the environment based on the reasoning/planning and the feedback it perceives over time. LLM agent has achieved impressive successes in embodied AI (Ahn et al., 2022; Huang et al., 2022a; Wang et al., 2023a), natural science (Wu et al., 2023; Swan et al., 2023), and social science (Park et al., 2022; 2023) applications.

Besides being *dynamic*, another increasingly captivating feature of LLM-based decision-making is the involvement of *strategic* interactions, oftentimes among multiple LLM agents. For example, it has been reported that the reasoning capability of LLMs can be improved by interacting with each other through negotiation and/or debate games (Fu et al., 2023; Du et al., 2023); LLM agents have now been widely used to *simulate* the strategic behaviors for social and economic studies, to understand the emerging behaviors in interactive social systems (Aher et al., 2023; Park et al., 2023). Moreover, LLMs have also exhibited remarkable potential in solving various games (Bakhtin et al., 2022; Mukobi et al., 2023), and in fact, a rapidly expanding literature has employed *repeated games* as a fundamental benchmark to understand the strategic behaviors of LLMs (Brookins & DeBacker,

---

[⋆]Equal contribution.

2023; Akata et al., 2023; Fan et al., 2023). These exciting empirical successes call for a rigorous examination and understanding through a theoretical lens of decision-making.

*Regret*, on the other hand, has been a core metric in (online) decision-making. It measures how "sorry" the decision-maker is, in retrospect, not to have followed the best prediction in hindsight (Shalev-Shwartz, 2012). It provides not only a sensible way to *evaluate* the sophistication level of online decision-makers, but also a quantitative way to measure their *robustness* against arbitrary (and possibly adversarial) environments. More importantly, it inherently offers a connection to modeling and analyzing *strategic behaviors*: the long-run interaction of no-regret learners leads to certain *equilibrium* when they repeatedly play games (Cesa-Bianchi & Lugosi, 2006). In fact, *no-regret* learning has served as a natural model for predicting and explaining human behaviors in strategic decision-making, with experimental evidence (Erev & Roth, 1998; Nekipelov et al., 2015; Balseiro & Gur, 2019). It has thus been posited as an important model of "rational behaviors" in playing games (Blum et al., 2008; Roughgarden, 2015). Hence, it is natural to ask:

> *Can we examine and better understand the online and strategic decision-making behaviors of LLMs through the lens of* regret*?*

Acknowledging that LLM(-agents) are extremely complicated to analyze, to gain some insights into the question, we focus on benchmark decision-making settings: online learning with convex (linear) loss functions, and playing repeated games. We defer a detailed literature review to Appendix A, and summarize our contributions as follows.

**Contributions.** First, we carefully examine the performance of several representative pre-trained LLMs in the aforementioned benchmark online decision-making settings, in terms of *regret*. We observe that LLM agents can achieve regret sublinear in time in (non-stochastic) online learning settings, where the loss functions change over time either arbitrarily, or by following some patterns with bounded variation, and in playing both representative and randomly generated repeated games. For the latter, equilibria will emerge as the long-term behavior of the multi-LLM interactions. Second, we provide some theoretical insights into the observed sublinear regret behaviors, based on certain assumptions on the *supervised pre-training* procedure, a common practice in training large models for decision-making, and some hypothetical models for training data generation. In particular, we make a connection of the pre-trained LLMs to the known no-regret algorithm of *follow-the-perturbed-leader* (FTPL) under these assumptions. Third, we also identify (simple) cases where advanced LLMs such as GPT-4 fail to be no-regret. We thus propose a novel *unsupervised* training loss, *regret-loss*, which, in contrast to the supervised pre-training loss, does not require the *labels* of (optimal) actions. We then establish both *statistical* and *optimization* guarantees for regret-loss minimization, which, in particular, show that minimizing such a loss can *automatically* lead to the known no-regret learning algorithm of *follow-the-regularized leader* (FTRL), under single-layer self-attention parameterization. Our further experiments demonstrate the effectiveness of our new loss, especially in addressing the above "regrettable" cases. With the fast development of LLMs, we emphasize that our goal is not to assert whether (current) LLMs are no-regret learners or not, especially given both the positive and negative observations above. Instead, our hope is to introduce and inspire more rigorous metrics and principles into the current evaluation and development of LLM agents, for online and multi-agent strategic decision-making.

## 2 PRELIMINARIES

**Notation.** For a finite set $\mathcal{S}$, we use $\Delta(\mathcal{S})$ to denote the simplex over $\mathcal{S}$. We denote $\mathbb{R}^+ := \{x \in \mathbb{R} \mid x \geq 0\}$. We define $\mathbf{0}_d$ and $\mathbf{1}_d$ as the $d$-dimensional all-zero and all-one vector, respectively, and $\boldsymbol{O}_{d \times d}$ and $I_{d \times d}$ as the $d \times d$-dimensional zero matrix and identity matrix, respectively. For a positive integer $d$, we define $[d] = \{1, 2, \ldots, d\}$. For $p \in \mathbb{R}^d, R > 0$ and $C \subseteq \mathbb{R}^d$ being a convex set, define $B(p, R, \|\cdot\|) := \{x \in \mathbb{R}^d \mid \|x - p\| \leq R\}$ and $\texttt{Proj}_{C,\|\cdot\|}(p) = \arg\min_{x \in C} \|x - p\|$. For any $x \in \mathbb{R}^d$, define $\texttt{Softmax}(x) = \left(\frac{e^{x_i}}{\sum_{i \in [d]} e^{x_i}}\right)_{i \in [d]}$. For a vector $v \in \mathbb{R}^n$, we use $\|v\|_p$ to denote its $L_p$-norm, with $\|v\|$ denoting the $L_2$-norm by default. We define $\mathbb{1}(\mathcal{E}) = 1$ if some event $\mathcal{E}$ is true, and $\mathbb{1}(\mathcal{E}) = 0$ otherwise. For a random variable $X$, we use $\texttt{supp}(X)$ to denote its support.

### 2.1 ONLINE LEARNING & GAMES

**Online learning.** We consider the online learning setting where an agent interacts with the environment for $T$ rounds, by iteratively making decisions based on the feedback she receives. Specifically, at each time step $t$, the agent chooses her decision policy $\pi_t \in \Pi$ for some bounded domain $\Pi$, and after her commitment to $\pi_t$, a bounded loss function $f_t : \Pi \to [-B, B]$ for some constant $B > 0$ is chosen by the environment, potentially in an adversarial fashion. The agent thus incurs a

loss of $f_t(\pi_t)$, and will update her decision to $\pi_{t+1}$ using the feedback. We focus on the most basic setting where the agent chooses actions from a finite set $\mathcal{A}$ every round, which is also referred to as the *Experts Problem* (Cover, 1966; Vovk, 1990; Littlestone & Warmuth, 1994; Hazan, 2016), without loss of much generality (c.f. Appendix B.5 for a detailed discussion). In this case, $\Pi$ becomes the simplex over $\mathcal{A}$, i.e., $\Pi = \Delta(\mathcal{A})$, and $f_t(\pi_t) = \langle \ell_t, \pi_t \rangle$ for some loss *vector* $\ell_t \in \mathbb{R}^d$ that may change over time, where $d := |\mathcal{A}|$.

At time step $t \in [T]$, the agent may receive either the full vector $\ell_t$, or only the realized loss $\ell_{ta_t}$ (we sometimes also interchangeably write it as $\ell_t(a_t)$), the $a_t$th element of $\ell_t$, for some $a_t \sim \pi_t(\cdot)$, as feedback, which will be referred to as online learning with *full-information feedback*, and that with *bandit feedback*, respectively. The latter is also referred to as the *adversarial/non-stochastic bandit* problem in the multi-armed bandit (MAB) literature. Note that hereafter, we will by default refer to this setting that does *not* make any assumptions on the loss sequence $(\ell_t)_{t \in [T]}$ simply as *online learning*. Moreover, if the loss functions change over time (usually with certain bounded variation), we will refer to it as *non-stationary online learning* for short, whose bandit-feedback version is also referred to as the *non-stationary bandit* problem.

**Repeated games.** The online learning setting above has an intimate connection to game theory. Consider a normal-form game $\mathcal{G} = \langle N, \{\mathcal{A}_n\}_{n \in [N]}, \{r_n\}_{n \in [N]} \rangle$, where $N$ is the number of players, $\mathcal{A}_n$ and $r_n : \mathcal{A}_1 \times \cdots \times \mathcal{A}_N \to [-B, B]$ are the action set and the payoff function of player $n$, respectively. The $N$ players repeatedly play the game for $T$ rounds, each player $n$ maintains a strategy $\pi_{n,t} \in \Delta(\mathcal{A}_n)$ at time $t$, and takes action $a_{n,t} \sim \pi_{n,t}(\cdot)$. The *joint* action $a_t = (a_{1,t}, \cdots, a_{N,t})$ determines the payoff of each player at time $t$, $\{r_n(a_t)\}_{n \in [N]}$. From a single-player's (e.g., player $n$'s) perspective, she encounters an online learning problem with (expected) loss function $\ell_t := -\mathbb{E}_{a_{-n,t} \sim \pi_{-n,t}}[r_n(\cdot, a_{-n,t})]$ at time $t$, where $-n$ denotes the index for all the players other than player $n$. We will refer to it as the *game setting* for short, and use the terms of "agent" and "player" interchangeably hereafter. The key difference between online learning and repeated games is in their interaction dynamics: online learning involves an agent facing a potentially adversarial, changing environment (or sequence of loss functions), while in repeated games, agents interact by playing the same game repeatedly, which might be less adversarial when they follow specific learning algorithms.

## 2.2 Performance Metric: Regret

We now introduce *regret*, the core performance metric used in online learning and games. For a given algorithm $\mathscr{A}$, let $\pi_{\mathscr{A},t}$ denote the decision policy of the agent at time $t$ generated by $\mathscr{A}$. Then, the regret, which is the difference between the accumulated (expected) loss incurred by implementing $\mathscr{A}$ and that incurred by the best-in-hindsight fixed decision, can be defined as

$$\text{Regret}_{\mathscr{A}}\big((f_t)_{t \in [T]}\big) := \sum_{t=1}^{T} f_t(\pi_{\mathscr{A},t}) - \inf_{\pi \in \Pi} \sum_{t=1}^{T} f_t(\pi).$$

In the Experts Problem, the definition is instantiated as $\text{Regret}_{\mathscr{A}}((\ell_t)_{t \in [T]}) := \sum_{t=1}^{T} \langle \ell_t, \pi_{\mathscr{A},t} \rangle - \inf_{\pi \in \Pi} \sum_{t=1}^{T} \langle \ell_t, \pi \rangle$. With bandit-feedback, a common metric may also take further expectation for $\text{Regret}_{\mathscr{A}}$, over the randomness of the policies $(\pi_{\mathscr{A},t})_{t \in [T]}$. An algorithm $\mathscr{A}$ is referred to as being *no-regret*, if $\max_{(f_t)_{t \in [T]}} \text{Regret}_{\mathscr{A}}((f_t)_{t \in [T]}) \sim o(T)$, i.e., the worse-case regret grows sublinearly in $T$. Known no-regret algorithms include follow-the-regularized-leader (Shalev-Shwartz & Singer, 2007), follow-the-perturbed-leader (Kalai & Vempala, 2005) (see Appendix B.4 for more details).

In non-stationary online learning, one may also use the metric of *dynamic regret* (Zinkevich, 2003), where the *comparator* in the definition also changes over time, as the best decision policy at each time $t$: $\text{D-Regret}_{\mathscr{A}}((f_t)_{t \in [T]}) := \sum_{t=1}^{T} f_t(\pi_{\mathscr{A},t}) - \sum_{t=1}^{T} \inf_{\pi \in \Pi} f_t(\pi)$, which is a stronger notion than $\text{Regret}_{\mathscr{A}}((f_t)_{t \in [T]})$ in that $\text{Regret}_{\mathscr{A}}((f_t)_{t \in [T]}) \leq \text{D-Regret}_{\mathscr{A}}((f_t)_{t \in [T]})$.

## 3 Do Pre-Trained LLMs Have Regret? Experimental Validation

In this section, we explore the no-regret behaviors of representative LLMs (i.e., mainly GPT-4 Turbo and GPT-4, together with GPT-3.5 Turbo, Mixtral-8x7b-instruct, and Llama-3-70B-instruct), in the context of online learning and games. All experiments with LLMs are conducted using the public OpenAI (Openai, 2023) or LLM Engine (LLM Engine, 2023) Python API. We provide some hypothetical intuitions as to why pre-trained LLM might be no-regret in Appendix C.1, which will be made concrete next.

**Interaction protocol.** To enable the sequential interaction with LLMs, we first describe the setup and objective of our experimental study. At each round, we incorporate the entire history of loss vectors of past interactions into our prompts, as concatenated texts, and ask the LLM agent to determine

a policy that guides the decision-making for the next round. Note that since we hope to *evaluate* the sophistication level of pre-trained LLMs through online learning or games, we only provide simple prompts that she should utilize the history information, without providing explicit rules of *how* to make use of the history information, nor asking her to *minimize regret* (in any sense). A detailed description and an ablation study of the prompts are deferred to Appendix C.8, and an illustration of the protocol for playing repeated games is given in Figure C.1.

### 3.1 FRAMEWORK FOR SUBLINEAR REGRET BEHAVIOR VALIDATION

Before delving into the results, we note that to the best of our knowledge, we are not aware of any principled framework for validating sublinear growth of the regret with *finite-time* experimental data. Therefore, we propose two frameworks below to rigorously validate the no-regret behaviors of algorithms over a *finite $T$*, which might be of independent interest. More details can be found in Appendix C.3.

**Trend-checking framework.** We propose a statistical hypothesis test aligned with our objectives:

$$H_0 : \text{The sequence } \left(\text{Regret}_{\mathscr{A}}\left((f_\tau)_{\tau \in [t]}\right)/t\right)_{t \in [T]} \text{ does not exhibit a decreasing pattern}$$

$$H_1 : \text{The sequence } \left(\text{Regret}_{\mathscr{A}}\left((f_\tau)_{\tau \in [t]}\right)/t\right)_{t \in [T]} \text{ shows a decreasing pattern.}$$

Ideally, one should check if $\text{Regret}_{\mathscr{A}}\left((f_\tau)_{\tau \in [t]}\right)/t$ approaches zero (or a negative value) as $t$ goes to infinity. With a finite $T$ value, testing these hypotheses provides a method to quantify this – whether we reject $H_0$ offers a way to measure it. To this end, one needs to count the number of $\text{Regret}_{\mathscr{A}}\left((f_\tau)_{\tau \in [t]}\right)/t - \text{Regret}_{\mathscr{A}}\left((f_\tau)_{\tau \in [t+1]}\right)/(t+1) > 0$, for which we use Proposition 1 below. We will report the $p$-value of $H_0$, denoted as $p_{trend}$, as the output of this framework.

**Proposition 1.** (*$p$-value of the null hypothesis*)**.** *Define the event*

$$\mathcal{E}(s, T) := \left\{ \text{The number of } \frac{\text{Regret}_{\mathscr{A}}\left((f_\tau)_{\tau \in [t]}\right)}{t} - \frac{\text{Regret}_{\mathscr{A}}\left((f_\tau)_{\tau \in [t+1]}\right)}{t+1} > 0 \text{ for } t = 1, \ldots, T \text{ is at least } s \geq \frac{T-1}{2} \right\}.$$

*Under the assumption that the null hypothesis $H_0$ holds, the probability of this event happening is bounded as $\mathbb{P}_{H_0}(\mathcal{E}(s, T)) \leq \frac{1}{2^{T-1}} \sum_{t=s}^{T-1} \binom{T-1}{t}$.*

**Regression-based framework.** We propose an alternative approach by fitting the data with regression. In particular, one can use the data $\left\{ \left(t, \log \text{Regret}_{\mathscr{A}}\left((f_\tau)_{\tau \in [t]}\right)\right)\right\}_{t \in [T]}$ to fit a function $g(t) = \beta_0 \log t + \beta_1$, where the estimate of $\beta_0$, i.e., $\widehat{\beta}_0$, satisfying $\widehat{\beta}_0 < 1$ may be used to indicate the no-regret behavior, i.e., the *sublinear* growth of $\text{Regret}_{\mathscr{A}}\left((f_\tau)_{\tau \in [t]}\right)$ over time. While being simple, it cannot be directly used when $\text{Regret}_{\mathscr{A}}\left((f_\tau)_{\tau \in [t]}\right) < 0$. Hence, we set $\log \text{Regret}_{\mathscr{A}}\left((f_\tau)_{\tau \in [t]}\right)$ as $-10$ if this happens. We define $p_{reg}$ as the $p$-value of the regression parameter $\widehat{\beta}_0$, and will report the pair of $(\widehat{\beta}_0, p_{reg})$ as the output of this framework.

### 3.2 RESULTS: ONLINE LEARNING

We now present the experimental results of pre-trained LLMs in online learning in: 1) (arbitrarily) changing environments, 2) non-stationary environments, and 3) bandit-feedback environments. Results for 2) and 3) are deferred to Appendices C.4 and C.5.

**Changing environments.** We first consider the setting with (arbitrarily) changing environments, which are instantiated as follows: 1) *Randomly-generated loss sequences.* At every timestep, we generate a random loss vector $\ell_t \sim \text{Unif}(\times_{i=1}^d [\min\{x_i, y_i\}, \max\{x_i, y_i\}])$ for $\{x_i, y_i \sim \text{Unif}(0, 10)\}_{i \in [d]}$ or $\ell_t \sim \mathcal{N}(\boldsymbol{\mu}_d, I)$ with clipping to $[0, 10]$ to ensure boundedness of the loss, where $\boldsymbol{\mu}_d \sim \text{Unif}([0, 10]^d)$. Note that we use this as a way to *systematically* generate potentially arbitrary loss sequences, and also note that our regret was defined for each *realization* of the *random loss vectors* (instead of their expectations as in the definition of regret in *stochastic bandit* problems), which can be arbitrarily different across timesteps. 2) *Loss sequences with certain trends.* Although many real-world environments may change, they often change by following certain patterns. Therefore, we consider two representative trends, the *linear* trend and the *periodic* (sinusoid) trend. We sample $a, b \sim \text{Unif}([0, 10]^d)$ and let $\ell_t = (b - a)\frac{t}{T} + a$ for the linear trend and $\ell_t = 5(1 + \sin(at + b))$ for the periodic trend. In the experiments, we choose $d = 2$. The average regret (over multiple randomly generated instances) performance is presented in Figure 3.1[1], where we compare GPT-4 with well-known no-regret algorithms, FTRL with entropy regularization and FTPL with Gaussian perturbations (with tuned parameters). It is seen that these pre-trained LLMs can achieve sublinear regret in a large portion of the instances, and have sometimes even lower regret values than baselines.

---

[1]We emphasize that the error bars in the figures are *not* associated with the randomness/variance of the *algorithms/LLM-agents*, but with the randomness/variance of the *generation of environment instances*.

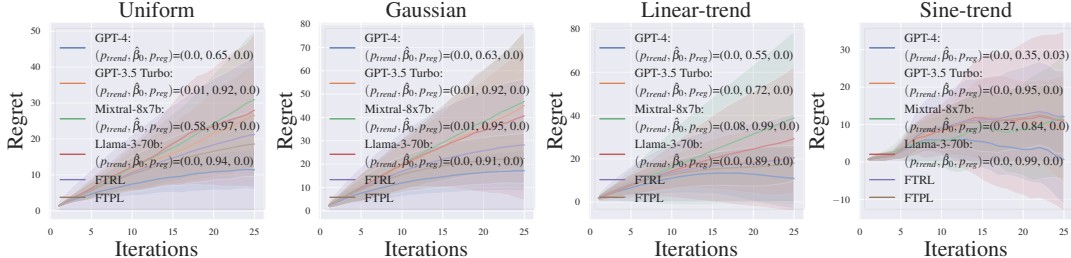

Figure 3.1: Regret of pre-trained LLMs for online learning with full-information feedback. Notably, both commercial and open-source LLMs can achieve sublinear regret as validated by our frameworks and the comparison with FTRL/FTPL, though the performances of weaker models of GPT-3.5 and open-source ones are worse. Interestingly, the GPT-4 model can even outperform well-known no-regret learning algorithms, FTRL and FTPL.

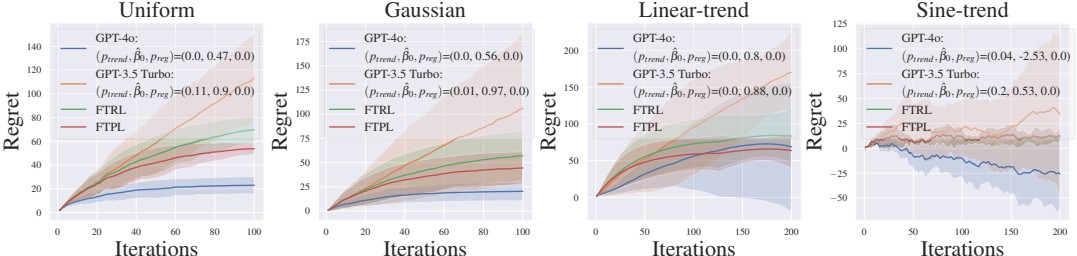

Figure 3.2: Regret of pre-trained LLMs for online learning with full-information feedback, with longer horizons of $T = 100$ and $T = 200$. In most cases, the LLMs can achieve sublinear regret as validated by our frameworks and the comparison with FTRL/FTPL, though the performances of the weaker model of GPT-3.5 is worse.

**Behavioral patterns of LLMs.** To understand how LLMs make decisions at each time step, we provided example outputs of LLMs *reasoning* how they generate their policies in Appendix C.10. We find that LLMs tend to use the history of the reward vectors by looking at their *sum/average*, and tend to introduce *randomization* in decision-making. These are known to be the keys to achieving no-regret behaviors in online learning (Hazan, 2016; Cesa-Bianchi & Lugosi, 2006).

**Longer-horizon results.** We also test the robustness and scalability of our empirical findings in more challenging environments. We extend the problem horizon to $T = 100$ for the two settings where loss vectors are generated in a stationary way (i.e., *Uniform* and *Gaussian*), and $T = 200$ for the other two non-stationary settings (i.e., *Linear-trend* and *Sine-trend*). Note that since in each round, we need to feed all the previous history to the LLMs, the API costs in fact scale *quadratically* with respect to the horizon $T$. Therefore, we replace GPT-4 by its cheaper (and more recent) version of GPT-4o. To further scale to even longer-horizon cases with $T = 500$, we *summarize* the history to reduce the prompt length by providing LLMs with the summation of the history loss associated with each action. Similar summary-based input was also used in the concurrent work Krishnamurthy et al. (2024), where both the *averaged reward* and the *action selection count* of each action were summarized for the (i.i.d.) stochastic bandit setting. The corresponding results are provided in Figure 3.2 and Table 1, where the LLMs can exhibit no-regret behaviors as validated by our frameworks and the comparison with FTRL/FTPL.

| $(p_{trend}, \widehat{\beta}_o, p_{reg})$ | GPT-4o | FTRL | FTPL |
|---|---|---|---|
| Uniform | (0.0, 0.85, 0.0) | (0.0, 0.6, 0.0) | (0.0, 0.52, 0.0) |
| Gaussian | (0.0, 0.86, 0.0) | (0.0, 0.64, 0.0) | (0.0, 0.68, 0.0) |
| Linear-trend | (0.02, 0.83, 0.5) | (0.02, 0.76, 0.1) | (0.01, 0.79, 0.0) |
| Sine-trend | (0.09, 0.28, 0.0) | (0.01, 0.24, 0.0) | (0.01, 0.26, 0.0) |

Table 1: Longer-horizon ($T = 500$). GPT-4o model can still exhibit sublinear regret behaviors as validated by our frameworks and the comparison with FTRL/FTPL.

### 3.3 RESULTS: MULTI-PLAYER REPEATED GAMES

We now consider the setting when multiple LLMs make online decisions in a *shared* environment repeatedly. Specifically, at each round, the loss vectors each agent receives are determined by both her payoff matrix and the strategies of all other agents. Note that the payoff matrix is not directly

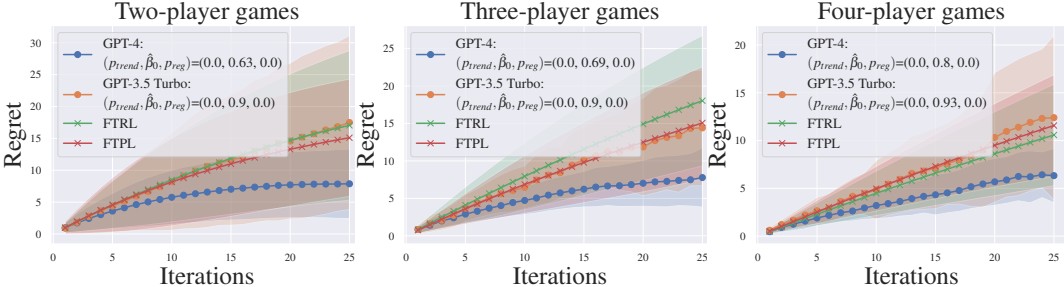

Figure 3.3: Regret of pre-trained LLMs for repeated games of different sizes, n most cases, both commercial and open-source LLMs can achieve sublinear regret as validated by our frameworks and the comparison with FTRL/FTPL. We report the regret of one agent for ease of presentation.

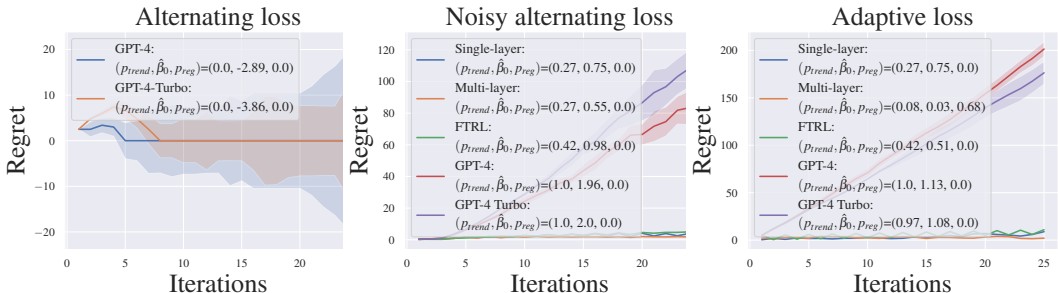

Figure 3.4: (left) Regret of GPT-4 (Turbo) under the canonical counterexample for FTL (Hazan, 2016, Chapter 5). (mid, right) Failure of GPT-4 (Turbo) on two scenarios with regrettable behaviors, while Transformers trained by our new regret-loss ($N = 1$) in Section 5 can achieve sublinear regret.

revealed to the LLM agent, but she has to make decisions in a completely online fashion based on the payoff vector marginalized by the opponents' strategies (see Figure C.1 for an example of the prompt). This is a typical scenario in learning in (repeated) games (Cesa-Bianchi & Lugosi, 2006).

**Representative games.** We first test LLMs on 6 representative general-sum games (*win-win, prisoner's dilemma, unfair, cyclic, biased,* and *second best*) studied in Robinson & Goforth (2005) (c.f. Appendix B.6). For each type of the game, we conduct 20 repeated experiments.

**Randomly generated games.** To further validate the no-regret behaviors of LLMs, we also test on 50 randomly generated three-player general-sum games, and 50 randomly generated four-player general-sum games, where each entry of the payoff matrix is sampled randomly from $\text{Unif}([0, 10])$. These are larger and more challenging settings than the structured and representative ones above.

We summarize the experimental results in Figure 3.3, which are similar to the above in the online setting: for all types of games, pre-trained LLMs can achieve sublinear regret, which is often lower than that obtained by FTRL/FTPL for most games. We provide six instances of three-player general-sum games and six instances of four-player general-sum games in Figure C.4 and Figure C.5, respectively. Occasionally, GPT-4 even provides a negative regret value.

### 3.4 PRE-TRAINED LLM AGENTS CAN STILL HAVE REGRET

The experiments above may suggest the no-regret behaviors of LLMs in online learning and game playing. However, is this capability *universal*? We show that the no-regret property can break for LLM agents if the loss vectors are generated in a more adversarial way.

**Canonical counterexamples for follow-the-leader.** First, we consider two well-known examples that the *follow-the-leader* (FTL) algorithm (Shalev-Shwartz, 2012) suffers from *linear regret*.

***Example 1:*** $\ell_1(1) = 5, \ell_1(2) = 0$ and $\ell_t(2 - t\%2) = 10, \ell_t(1 + t\%2) = 0$ for $t \geq 2$ (Hazan, 2016).

***Example 2:*** $\ell_t(2 - t\%2) = 10, \ell_t(1 + t\%2) = 0$ for $1 \leq t \leq c$ and $\ell_t(1) = 10, \ell_t(2) = 0$ for $c + 1 \leq t \leq T (= 500)$, for some integer $c$ satisfying $0 < c < T$ (Feder et al., 1992).

Here, $\%$ denotes the modulo operation. Interestingly, for ***Example 1***, GPT-4 agent can easily identify the pattern for the loss sequence that the optimal action *alternates*, thus accurately predicting the loss it will receive and achieving low regret in Figure 3.4. For ***Example 2***, the GPT-4 agent with *raw history* input also provides an impressively lower (negative) regret than FTRL and FTPL

(Figure C.6). The GPT-4 agent with *summarized history* input, in contrast, suffers from much larger regret than FTRL and FTPL. We defer the detailed comparison between using raw history and summarized history to Figure C.6, and an explanation of LLMs' behaviors via predicting the *trend* of the loss instances to Appendix C.7. In summary, the GPT-4 agent may predict such worst-case sequences well, and does not fail in the same way as FTL, which is known to suffer from a lack of randomness in decisions.

Additionally, the results on ***Example 2*** also imply that summary-based history input can perform worse than the raw-history-based one in the adversarial setting we consider, while the former was claimed to be the key in succeeding in the i.i.d. stochastic bandit setting (Krishnamurthy et al., 2024). The regret values with these two types of input differ significantly, with a *p*-value of $1.2 \times 10^{-157}$ under a one-sided independent t-test. These results further illustrate the fundamental differences between the settings considered in Krishnamurthy et al. (2024) and ours.

**Noisy alternating loss sequence.**  Inspired by the above, we design a new loss sequence that is *similar but less predictable*, by adding some noise to the canonical counterexample. Specifically, we construct the following (simple) loss sequence with 2 actions such that $\ell_t(1 + t\%2) = \min(25/t, 10), \ell_t(2 - t\%2) \sim \text{Unif}([9, 10])$ for $t \in [25]$.

**Adaptive loss sequence.**  We also develop a simpler but more *adaptive* loss sequence that takes the full power of the adversary in our online learning setup. After the GPT-4 agent provides $\pi_t$, we choose $\ell_t$ with $\ell_t(\arg\max_i \pi_{ti}) = 10$ and $\ell_t(3 - \arg\max_i \pi_{ti}) = 0$.

We also report the average regret over 20 repeated experiments for the later two settings using GPT-4 and more advanced GPT-4 Turbo in Figure 3.4, where we cannot reject the hypothesis that GPT-4 (Turbo) has linear regret by either our trend-checking or regression-based framework. These observations have thus motivated us to design new approaches to further promote the no-regret behaviors of the models, with additional training, as to be detailed in Section 5. Before it, we first provide some theoretical insights into the observed sublinear regret behaviors.

## 4 WHY DO PRE-TRAINED LLMS (NOT) HAVE REGRET? A HYPOTHETICAL MODEL AND SOME THEORETICAL INSIGHTS

We now provide some plausible explanations about the observed no-regret behaviors of pre-trained LLMs, which are highly *hypothetical* by nature, since to the best of our knowledge, the details of pre-training these popular LLMs (e.g., GPT-3.5 Turbo and GPT-4), regarding data distribution, training algorithm, etc., have not been revealed. We instead make the explanations based on some existing assumptions in the literature for modeling human behaviors, and the recent literature on understanding LLMs and Transformers.

### 4.1 A (HUMAN) DECISION-MAKING MODEL: QUANTAL RESPONSE

A seminal model for human decision-making behaviors is the *quantal response* model, which assumes that humans are often imperfect decision-makers, and their *bounded rationality* can be modeled through unseen *latent variables* that influence the decision-making process (McFadden, 1976; McKelvey & Palfrey, 1995), for which we defer the formal definition and introduction to Appendix D.2. In online decision-making, given the *history* information with *multiple* loss vectors, we adopt the following generalization of the quantal response model.

**Definition 4.1** (Quantal response against multiple losses). *Given a set of losses* $(\ell_i)_{i \in [t]}$, *a noise distribution* $\epsilon \sim P_{noise}$, *and* $\eta_t > 0$, *the generalized quantal response against* $(\ell_i)_{i \in [t]}$ *is defined as*

$$P_{quantal}^{\eta_t}\left(a \mid (\ell_i)_{i \in [t]}\right) := P_{quantal}^{\eta_t}\left(a \,\middle|\, \sum_{i=1}^{t} \ell_i\right) = \mathbb{P}\left(a \in \arg\min_{a' \in \mathcal{A}} \ z(a')\right), \text{ where } z = \eta_t \epsilon + \sum_{i=1}^{t} \ell_i.$$

In simpler terms, the generalized quantal response is defined as the standard quantal response against the *summation* of the losses. Such a model has been investigated in the learning-in-games and behavioral economics literature (see Appendix D.2 for more details). Such a definition is also aligned with our empirical findings on LLMs' behavioral patterns in Section 3.2: i) evaluating the summation/average; ii) introducing randomization in decision-making. To gain more insights into these empirical findings, we next analyze a case where pre-training under certain assumptions provably leads to the quantal response behaviors and further yields no-regret guarantees.

### 4.2 CASE STUDY: PRE-TRAINING UNDER CANONICAL DATA DISTRIBUTION

Pre-training of LLMs is predominantly based on *next-token prediction*. When applying LLMs to sequential decision-making, the model receives the context of the decision-making task as

$(x_1, x_2, \cdots, x_N)$ and then generates $(x_{N+1}, \cdots, x_M)$ encoding the *action* for some $N, M \in \mathbb{N}^+$ and $N < M$, where each $x_i \in \mathcal{V}$ represents one *natural language token* for $i \in [M]$, and $\mathcal{V}$ is the finite token set. This process can be conceptualized as *predicting the optimal action* in the form of the next token prediction (Yao et al., 2023b; Shinn et al., 2023; Liu et al., 2023a;e). Note that this training procedure may also appear in the form of *supervised fine-tuning (SFT)* for downstream tasks of decision-making or question-answering, where optimal action labels may be easier to obtain (Cobbe et al., 2021; Li et al., 2022; Lewkowycz et al., 2022). Meanwhile, large models are often (pre-)trained under several *fixed/stationary* environments (Laskin et al., 2023; Lin et al., 2024; Lee et al., 2023; Reed et al., 2022), which may limit their ability to handle *arbitrary/non-stationary/adversarial* loss sequences in online learning. Thus, it is natural to ask: *Is it possible to have no-regret behaviors emerging as a consequence of this (optimal) action prediction, under only a* fixed *pre-training distribution of the environments?*

Here we analyze a standard pre-training objective on a token sequence distribution $x_{1:N_{t+1}} \sim P_t^{text}$ for given $t \in [T]$, which is the expected log-likelihood maximization for next-token prediction over $\Theta$, the parameter space of the LLM:

$$\max_{\theta \in \Theta} \quad \mathbb{E}_{x_{1:N_{t+1}} \sim P_t^{text}} \sum_{j=1}^{N_{t+1}} \log \mathrm{LLM}_\theta \left( x_j \,|\, x_{1:j-1} \right), \qquad (4.1)$$

where we define $\mathrm{LLM}_\theta \left( x_1 \,|\, x_{1:0} \right) = \mathrm{LLM}_\theta \left( x_1 \right)$.

For the pre-training distribution, we model it as follows: there exists a latent variable $z$, representing the loss for the underlying *static* decision-making problem. The pre-training dataset, however, only contains *partial observations* $x_{1:N_t}$ (a natural language representation of $\ell_{1:t}$) of $z$ due to imperfect data collection, which could be attributed to the fact that $z$ is private to the data-generator (human), representing the actual intention of the human/data-generator. Hence, LLM will only be pre-trained with partial and noisy information about $z$. Meanwhile, we assume that some high-quality action label $x_{N_t+1:N_{t+1}}$ (a natural language representation of $a$) with respect to the underlying loss vector $z$ is also available in the dataset, which could come from user surveys, personal blogs, or data annotation. We formalize such an assumption:

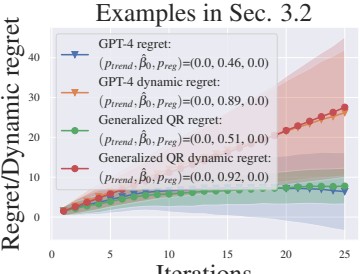

Figure 4.1: Comparison of GPT-4 with the generalized QR model, where the model can very well capture the behavior of the GPT-4 agent for examples in Section 3.2.

**Assumption 1** (Pre-training distribution). *Given* $T \in \mathbb{N}^+$, $t \in [T]$, $N_{t+1} \in \mathbb{N}^+$, *there are latent variables* $(z, \ell_{1:t})$, $N_1, \cdots, N_t \in [N_{t+1}]$, $N_0 = 0$, *such that* $\mathbb{P}(z, \ell_{1:t}, x_{1:N_{t+1}}) = \mathbb{P}(z, \ell_{1:t})\mathbb{P}(x_{1:N_t} \,|\, \ell_{1:t})\mathbb{P}(x_{N_t+1:N_{t+1}} \,|\, z)$, *and* $P_t^{text}(x_{1:N_{t+1}}) := \mathbb{P}(x_{1:N_{t+1}}) = \int_z \int_{\ell_{1:t}} \mathbb{P}(z, \ell_{1:t}, x_{1:N_{t+1}})d\ell_{1:t}dz$. *Intuitively, tokens* $\{x_{N_{i-1}+1:N_i}\}_{i \in [t]}$ *encode the context, i.e., information for* $\ell_{1:t}$, *and the user will decode action* $a$ *from* $x_{N_t+1:N_{t+1}}$.

To further understand our assumption, we provide an example in Appendix D.3, showing how a natural text corpus may satisfy it. Similar assumptions that suppose the existence of such latent variables in generating the pre-training datasets have also been made recently in Lee et al. (2023); Lin et al. (2024); Liu et al. (2023e), for understanding the in-context decision-making behaviors of LLMs/Transformers through posterior sampling, for which we defer a detailed comparison to Appendix D.8. In particular, we show in Theorem 4.1 that if the noise, i.e., $\ell_i - z$ is modeled as Gaussian distributions and $x_{N_t+1:N_{t+1}}$ encodes the optimal action for $z$, the pre-trained LLM provably recovers the prominent human behavior model in Section 4.1, the quantal response model.

**Theorem 4.1** (Informal: Emergence of no-regret behavior). *Suppose Assumption 1 holds with both the prior distribution of* $z$ *and the conditional distribution of* $\{\ell_i \,|\, z\}_{i \in [t]}$ *being Gaussian, and* $x_{N_t+1:N_{t+1}}$ *encodes the optimal action for* $z$. *Then, with the function class of* $\mathrm{LLM}_\theta$ *being expressive enough, and* $\theta^\star$ *being a maximizer of Equation* (4.1), *the behavior of* $\mathrm{LLM}_{\theta^\star}$ *follows Definition 4.1. Furthermore, the use of* $\mathrm{LLM}_{\theta^\star}$ *can achieve no (dynamic) regret for (non-stationary) online learning with full-information/bandit feedback for arbitrary loss vectors (with bounded variation).*

The formal statement and proof are deferred to Appendix D.6. The results show that even when pre-training is conducted solely with loss vectors generated from *stationary* distributions ($\ell_{1:t}$ are i.i.d. conditioned on $z$), it can still enable the *emergence of no-regret behaviors* in online learning against *potentially adversarial* losses. Key in the proof is the connection of pre-trained LLM models to the online learning algorithm of FTPL. Furthermore, Assumption 1 can be relaxed to better match

the actual LLMs' pre-training data distributions from diverse sources (c.f. Appendix D.7), and the prior distribution of $z$ could also be replaced by a general distribution (c.f. Theorem D.2). Finally, we point out its implications for playing games in Appendix D.6.1.

**How well can our hypothetical model class predict actual LLMs' behaviors?** To further verify our theoretically-justified model in Theorem 4.1, we propose to *estimate* the parameters of $\{\eta_t\}_{t=0}^{T-1}$ in Definition 4.1 using the interaction data with actual LLMs, and use the estimated model to predict LLMs' behaviors on some test set. In Figure 4.1, we show the averaged regret for the LLMs and our estimated model, where the generalized quantal response can *very well capture* the behavior of the LLM agent for all problem instances in Section 3.2, on which the LLMs oftentimes achieve sublinear regret, justifying the applicability of our hypothetical model and assumptions.

Finally, we acknowledge that for existing pre-trained LLMs like GPT-4, the canonical assumptions above, though may be further relaxed (c.f. Remark D.3), may not hold in general. More importantly, the *supervision labels*, i.e., the optimal action given $z$, may be sometimes imperfect or unavailable in the dataset. These caveats motivate the study in our next section.

## 5  PROVABLY PROMOTING NO-REGRET BEHAVIOR BY A NEW LOSS

In light of the observations in Section 3, we ask the question:

*Is there a way to enhance the no-regret property of the models **without** (optimal) action labels?*

To address this question, we propose to train models with a new *unsupervised learning* loss that naturally provides no-regret behaviors. We will particularly focus on the *Transformer* architecture (Vaswani et al., 2017) under this new loss, a common architecture used in most existing LLMs.

### 5.1  A NEW UNSUPERVISED TRAINING LOSS: *Regret-Loss*

Intuitively, our new training loss is designed to enforce the trained models to minimize regret under an arbitrary sequence of loss vectors. Specifically, we define the training loss as

$$\mathcal{L}(\theta) := \max_{\ell_1,\dots,\ell_T} \quad \text{Regret}_{\text{LLM}_\theta}\big((\ell_t)_{t\in[T]}\big) \tag{5.1}$$

where $\|\ell_t\|_\infty \leq B$ for $t \in [T]$. As discussed in Kirschner et al. (2023), directly minimizing the max regret can be computationally challenging, except for superficially simple problems. Moreover, Equation (5.1) is not necessarily differentiable with respect to the parameter $\theta$, if it does not satisfy the condition of Danskin's Theorem (Danskin, 1966); or even if it is differentiable (i.e., the maximizer of $(\ell_t)_{t\in[T]}$ is unique), computation of derivatives can be challenging since we need to calculate $\arg\max_{(\ell_t)_{t\in[T]}} \text{Regret}_{\text{LLM}_\theta}((\ell_t)_{t\in[T]})$ while there is an $\inf$ in the definition of regret. Therefore, we provide a general class of surrogate losses to approximate Equation (5.1):

$$\mathcal{L}(\theta,k,N) := \mathbb{E}\left[\frac{\sum_{j\in[N]} h(\text{Regret}_{\text{LLM}_\theta}((\ell_t^{(j)})_{t\in[T]})) f(\text{Regret}_{\text{LLM}_\theta}((\ell_t^{(j)})_{t\in[T]}),k)}{\sum_{j\in[N]} f(\text{Regret}_{\text{LLM}_\theta}((\ell_t^{(j)})_{t\in[T]}),k)}\right], \tag{5.2}$$

where $k \in \mathbb{N}^+$, $N \in \mathbb{N}^+$, $h : \mathbb{R} \to \mathbb{R}^+$ is a continuous function, with continuous derivative $h'$, and $f(\cdot,k) : \mathbb{R} \to \mathbb{R}^+$ is a continuous function for each $k \in \mathbb{N}^+$, satisfying $\lim_{k\to\infty} \frac{f(R_1,k)}{f(R_2,k)} = \infty \cdot \mathbb{1}(R_1 > R_2) + \mathbb{1}(R_1 = R_2)$, where we use the convention of $\infty \cdot 0 = 0$. These conditions on $h, f$ will be assumed throughout the paper. Examples of such an $f$ include $f(x,k) = x^k$ and $\exp(kx)$. We will sample $N$ trajectories of loss sequences $(\ell_t^{(j)})_{t\in[T],j\in[N]}$ from some continuous probability distribution supported on $[-B,B]^{T\times N}$ (without other additional statistical assumptions), and the expectation in Equation (5.2) is thus taken with respect to this distribution. In Appendix E.2, we prove that under certain regularity conditions of $f$ and $h$, we have

$$\lim_{N,k\to\infty} \mathcal{L}(\theta,k,N) = h\left(\max_{\ell_1,\dots,\ell_T} \text{Regret}_{\text{LLM}_\theta}((\ell_t)_{t\in[T]})\right),$$

and the uniform convergence of $\mathcal{L}(\theta,k,N)$: $\lim_{N,k\to\infty} \sup_{\theta\in\Theta} \Big| h\left(\max_{\ell_1,\dots,\ell_T} \text{Regret}_{\text{LLM}_\theta}((\ell_t)_{t\in[T]})\right) - $

$\mathcal{L}(\theta,k,N)\Big| = 0$, where $\Theta$ is a compact set of the model parameters. Hence, one can expect that minimizing the loss function in Equation (5.2) with large enough $k$ and $N$ may promote the trained models to have a small regret value. We will hereafter refer to Equation (5.2) as the *regret-loss*.

## 5.2 GENERALIZATION AND REGRET GUARANTEES OF REGRET-LOSS MINIMIZATION

We first establish a *statistical* guarantee under general parameterizations of $\text{LLM}_\theta$ that are Lipschitz with respect to $\theta$, including the Transformer-based models as used in GPT-4 and most existing LLMs (see Proposition 2 for an example with a formal statement). This guarantee focuses on their *generalization ability* when trained to minimize the empirical regret loss (c.f. Equation (E.3)), denoted as $\widehat{\mathcal{L}}(\theta, k, N, N_T)$, by replacing the expectation $\mathbb{E}$ in Equation (5.2) with the empirical mean using $N_T$ samples. We denote $\widehat{\theta}_{k,N,N_T} \in \arg\min_{\theta \in \Theta} \widehat{\mathcal{L}}(\theta, k, N, N_T)$, and present the generalization guarantee in Theorem E.1. Thanks to the uniform convergence of $\mathcal{L}(\theta, k, N)$ (c.f. Appendix E.2), we further obtain the following theorem on the regret guarantee of $\text{LLM}_{\widehat{\theta}_{k,N,N_T}}$:

**Theorem 5.1.** (Regret). *Suppose[2] for any $k \in \mathbb{N}^+$, $h, f(\cdot, k)$ are non-decreasing, and $\log f$ is a supermodular function (i.e., $\log f(R_1, k_1) - \log f(R_1, k_2) \geq \log f(R_2, k_1) - \log f(R_2, k_2)$ for $R_1 \geq R_2$ and $k_1 \geq k_2$). Then, with high probability, we have*

$$h\left(\lim_{N \to \infty} \lim_{k \to \infty} \max_{\|\ell_t\|_\infty \leq B} \text{Regret}_{\text{LLM}_{\widehat{\theta}_{k,N,N_T}}}\left((\ell_t)_{t \in [T]}\right)\right) \leq h\left(\inf_{\theta \in \Theta} \max_{\|\ell_t\|_\infty \leq B} \text{Regret}_{\text{LLM}_\theta}\left((\ell_t)_{t \in [T]}\right)\right) + \widetilde{\mathcal{O}}\left(\sqrt{\frac{d_\theta}{N_T}}\right).$$

We defer the proof of the theorem to Appendix E.4. Therefore, if additionally, the model parameterization (e.g., Transformers) can *realize* a no-regret algorithm (as to be shown next), then Theorem 5.1 means that with a large enough $N_T$, the learned $\text{LLM}_{\widehat{\theta}_{k,N,N_T}}$ becomes a *no-regret* learner, i.e., $\text{Regret}_{\text{LLM}_{\widehat{\theta}_{k,N,N_T}}}\left((\ell_t)_{t \in [T]}\right) = o(T)$. Finally, as a consequence, it is folklore that when multiple such LLMs interact, a coarse correlated equilibrium will emerge in the long-term (c.f. Corollary 1).

## 5.3 REGRET-LOSS TRAINED TRANSFORMERS CAN BE ONLINE LEARNING ALGORITHMS

Despite the generality of the previous results, one cannot use an *infinitely large* $N$ and $k$ in practice. Hence, we now provide results when $N$ is finite, for the architecture of *Transformer* models (Vaswani et al., 2017). We focus on single-layer (linear) self-attention models, as in most recent theoretical studies of Transformers (Ahn et al., 2023; Zhang et al., 2023a; Mahankali et al., 2023), and $N = 1$. Note that in this case, the choice of $f$ (and thus $k$) is not relevant. Thus, throughout this subsection, we drop superscript $(j)$ in Equation (5.2). We sample $\ell_t$ for $t \in [T]$ as realizations of some random variable $Z$, where we assume that $Z$ is symmetric about zero, and $\text{Var}(Z) = \Sigma \succ 0$. We consider the single-layer *linear* self-attention model as follows, for which we can show that the *global optimizer* of our regret-loss can automatically lead to a no-regret learning algorithm:

$$g(Z_t; V, K, Q, v_c, k_c, q_c) = \sum_{i=1}^{t} (V\ell_i + v_c)\left((K\ell_i + k_c)^\intercal \cdot (Qc + q_c)\right). \quad (5.3)$$

**Theorem 5.2.** *Consider the policy space $\Pi = B(0, R_\Pi, \|\cdot\|)$ for some $R_\Pi > 0$. The configuration of a single-layer linear self-attention model in Equation (5.3) $(V, K, Q, v_c, k_c, q_c)$ such that $K^\intercal(Qc + q_c) = v_c = \mathbf{0}_d$ and $V = -2R_\Pi \Sigma^{-1} \mathbb{E}\left(\|\sum_{t=1}^{T} \ell_t\|_1 \ell_1 \ell_2^\intercal\right) \Sigma^{-1}$ is a global optimal solution of Equation (5.2) with $N = 1$, $h(x) = x^2$. Moreover, every global optimal configuration of Equation (5.2) within the parameterization class of Equation (5.3) has the same output function $g$. Additionally, if $\Sigma$ is a diagonal matrix, then plugging any global optimal configuration into Equation (5.3), and projecting the output with $\text{Proj}_{\Pi, \|\cdot\|}$ is equivalent to FTRL with an $L_2$-regularizer.*

Theorem 5.2 not only shows the *capacity* of self-attention models: it can realize online learning algorithms, but also shows, more importantly, that minimizing our new regret-loss may *automatically* produce it. In particular, one does not need to hard-code the parameters of the Transformer to implement no-regret algorithms. Under single-layer self-attention parameterization (with softmax), we can also show that a *stationary point* of the loss function (Equation (5.2)) can lead to FTRL (c.f. Appendix E.5). Some potential generalizations of the results are also discussed in Appendix E.9.

## 5.4 EXPERIMENTAL RESULTS FOR REGRET-LOSS TRAINED TRANSFORMERS

We now provide experimental results for minimizing our *regret-loss* with the Transformer models, and evaluate in the following environments: 1) randomly-generated loss sequences (Figure E.3); 2) loss sequences with certain trends (Figure E.4); 3) repeated games (Figure E.5); and 4) counterexamples for pre-trained LLMs to be regrettable (Figure 3.4). Training setup can be found in Appendix E.11.1. We also provide an ablation study for optimizing Equation (5.2) in Appendix E.12.

Finally, we provide discussions on the limitations and future directions in Appendix F.

---

[2]Note that these conditions on $h, f$ are in addition to those specified after Equation (5.2).

## ACKNOWLEDGEMENT

The authors thank Constantinos Daskalakis, Kristian Georgiev, Noah Golowich, Dingwen Kong, Akshay Krishnamurthy, and Aleksander Madry for their helpful feedback. In particular, the authors thank Dingwen Kong for discussing the truncation idea in proving Lemma 8, and thank Akshay Krishnamurthy for bringing up a concurrent work that inspired our new experiments for the stochastic bandit setting that strengthened our paper. X.L. and K.Z. acknowledge the support from the U.S. Army Research Laboratory and the U.S. Army Research Office under grant number W911NF-24-1-0085 and NSF CAREER Award-2443704.

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

# Supplementary Materials for
# "Do LLM Agents Have Regret? A Case Study in Online Learning and Games"

CONTENTS

