# OpenReview forum: "Do LLM Agents  Have Regret? A Case Study in Online Learning and Games"
_ICLR.cc/2025/Conference — ICLR 2025 Poster_

### Official Review · Reviewer_KLuz · 2024-10-26

**Soundness:** 3
**Presentation:** 2
**Contribution:** 2
**Rating:** 6
**Confidence:** 2

**Summary:**

This paper investigates the performance of llm agents in online learning and game-theoretic scenarios. It empirically evaluates no-regret behaviors in pre-trained LLMs like GPT-4 across various online learning settings and repeated games, comparing them with classical no-regret algorithms. The authors introduce a new unsupervised training loss called "regret-loss," designed to enhance no-regret behavior without requiring action labels. It provides theoretical insights connecting pre-training to no-regret algorithms, and presents experimental results validating the proposed approaches.

**Strengths:**

1. It offers a theoretical framework for understanding no-regret behaviors in LLMs and introduces a novel training objective (regret-loss) with provable guarantees.
2. It includes a wide range of experiments with different synthetic & toy-case settings (arbitrary, non-stationary, and bandit feedback environments), comparing LLMs with established no-regret algorithms.

**Weaknesses:**

1. Misalignment of Framework and Problem: The application of online learning and game theory to evaluate LLM agents seems like over-engineering. LLMs are not explicitly designed for decision-making in the classical sense that these fields address (e.g., making repeated decisions based on feedback from an adversarial environment). Using regret minimization as a benchmark for LLM behavior could be seen as an attempt to impose a theoretical construct that may not provide meaningful insights into the actual strengths and weaknesses of LLMs.
2. Lack of Practical Relevance: If the focus is on understanding LLMs' interactive capabilities, other evaluation metrics might be more meaningful. For instance, metrics derived from natural language understanding or task-specific performance might offer more concrete insights into how LLMs perform as interactive agents. The notion of regret, which is central to online learning and game theory, may not be as intuitive or impactful in the context of LLMs used for dialogue, or other applications. There could be alternative frameworks that align better with evaluating and enhancing LLMs. For example, reinforcement learning, interactive learning environments, or even human-in-the-loop evaluations might offer more practical insights than regret-based analysis.

**Questions:**

1. How sensitive are the findings to the choice of synthetic games (win-win, prisoner’s dilemma, unfair, cyclic, biased, and second best) used for experiments?
2. How do the authors validate the importance of the research problem:

    "Can we examine and better understand the online and strategic decision-making behaviors of LLMs through the lens of regret"?

    The theoretical results, while rigorous, may feel detached from practical applications. The paper's findings on regret behavior might not translate into concrete benefits for LLM-based systems in the real world, potentially rendering the problem it addresses less relevant. This could make the work feel more like a theoretical exercise than a practical contribution to improving LLM capabilities.

---

> ### Author Response · Authors · 2024-11-21
> **Rebuttal by Authors (1)**
>
> We thank Reviewer KLuz for the detailed comments and insightful feedback. We address Reviewer KLuz's concerns and questions below:
>
> ## Regarding Misalignment of Framework and Problem
>
> We would like to clarify that the application of Online Learning and Game Theory to **evaluate/understand**  LLM agents (the main focus of our paper) is **not** over-engineering for the following reasons.
> - **Firstly, although LLMs were not trained for decision-making, employing LLMs for decision making in various applications are definitely not uncommon, and has achieved promising success**, see e.g., in medical decision-making [1], financial trading [2], recommendation systems [3, 4], and in fact, **"decision-making" is a core  domain "LLM-agents" are investigated and used for**.
> Many of the decision-making environments in those applications are indeed by nature *online*, *non-stationary*, and/or *strategic*. Therefore, since people have already been trying to deploy the LLMs in certain real-world applications that involve online or strategic decision-making, we believe our goal of **systematically evaluating and principally understanding LLMs through canonical online learning and game-theoretical tasks** is not over-engineering.
> - **Secondly, our work is not the only one that understands the online decision making ability of LLMs through numerical/synthetic problems in order to gain more insights/principles.** To name a few related prominent studies, [5, 6] studied  LLMs for multi-arm bandits, [7] studied LLMs for dueling bandits, and [8, 9] studied LLMs for game-theoretical problems.
> - **Finally, when evaluating LLMs in online learning and repeated games, we are not *engineering* them towards no-regret behaviors, but truthfully reporting its native performance to *understand* their capabilities/limits.** More importantly, we have shown positive results for a large portion of problems under such a setting. In other words, if we would have shown LLMs easily fail in some very challenging decision-making problems, we agree that such findings may sound trivial and letting LLMs solve such problems sounds like over-engineering since after all, LLMs are *not trained* for such problems. **Meanwhile, considering that LLMs already achieved promising success in many challenging online decision-making *applications*, systematically evaluating them through regret becomes quite fundamental and natural, since regret is probably the most important notion in Online Learning [12] and the key that bridges Online Learning and Game Theory [13].**
>
> ---
>
> ## Regarding Lack of Practical Relevance
>
> **We would like to emphasize that what we hope to evaluate in general is the *online decision-making capability* of LLMs, instead of its *natural language understanding/processing/generation ability*.** The necessity of understanding the decision-making ability is based on the empirical success LLMs have achieved in many synthetic or real-world **decision-making applications** as what we have stated before. Therefore, we believe it is more appropriate to utilize those metric designed to evaluate the quality of decision-making instead of those that evaluate the *quality of the natural languages/dialogues*.
>
> With our focus on the decision-making capability in mind, it is worth pointing out that the reason why we use regret is because it is a general enough metric that is widely applicable in various problems/environments, and is foundational in Online Learning and Game Theory. Specifically,
> - For reinforcement learning (in stochastic environments): Regret minimization is equivalent to accumulative rewards maximization. In fact, it is quite common in the RL theory literature to frame the objective of online RL as minimizing regret [10, 11], which leads to the optimal policy (through an online-to-batch argument).
> - For general interactive learning environments: as what we mentioned before, if the decision-making environment is interactive, no matter it is interacting with an unknown environment (i.e., the *online* setting in our paper), or unknown opponents (i.e., the *game* setting of our paper), regret is the most standard evaluation notion [12, 13]. ***More interestingly, regret has been shown to be a key factor that drives human interactive learning [14].*** Given the resemblance of LLMs to humans, the idea of understanding LLMs also through regret becomes quite natural.

---

> ### Author Response · Authors · 2024-11-21
> **Rebuttal by Authors (2)**
>
> ## Real-world case studies (added in Appendix C.11 of our revised paper)
>
> To further address the concern on the practical relevance, we have managed to perform case studies on two real-world settings as follows.
>
> As what we kindly reminded the reviewer before, we are **not** proposing a **new method to utilize LLMs** to solve **certain application tasks**, but rather to **evaluate** the rationality and **understand** the success of LLMs in sequential, non-stationary decision-making environments through **regret**. Given the focus of our paper, a meaningful evaluation on real-world case studies that could strengthen our message would be to validate whether our findings that LLMs can be no-regret still hold in some real-world case studies (although existing related works [5-8] that shared the same research focus have **only** done evaluation on **synthetic**  problems).
>
> ### Case Study 1) Sequential Recommendation.
>
> Towards such a goal, we consider the task of **sequential recommendation**, a task that people have been employing LLMs to solve with success. Note that how existing literature (e.g. [3, 4, 18]) uses LLMs to solve this task **exactly fits our Online Learning framework**, where humans feed a *history of items the user have interacted with* to the LLM, and then ask the LLM to *recommend the item* (or several items) the user may want to interact next, and the entire process carries on repeatedly. We use the real-world data and follow the experimental setup of [18], and  evaluate the regret of such an online learning process. We refer the detailed introduction for the new experiments and results to Appendix C.11 of our revised paper.
>
> From the results, we can see that LLMs are indeed no-regret **on such a real-world application with real-world data**. As a comparison, we also replace the real-world data with synthetic data generated in a uniformly random way, where  similar no-regret behaviors can still be observed. Also, interestingly, **a new insight that has come up is that LLMs even perform better on real-world data, which validates that real-world data may exhibit certain trend/structures, for which LLMs can exploit and thus, achieve superior performance, as we also showed in our paper.**
>
> ### Case Study 2) Interactive Negotiation.
>
>
> The experiment was designed to simulate negotiation scenarios between two agents, designated as LLM A and LLM B, over multiple turns. While previous studies have explored negotiation using LLMs (e.g.,  [19]), our setup differs by focusing on *language-based*   negotiation problems rather than *numerical* negotiation problems, which is more natural and closer to real-world applications. This shift allows for a more diverse exploration of negotiation scenarios, moving beyond purely quantitative frameworks.
>
> In each turn, a unique negotiation topic was generated using LLMs, providing context, objectives, and relevant background information. The negotiation is executed in a turn-based manner, where each turn consists of three steps: 1) Intention Generation: Each LLM defines its goal for the turn, specifying what it aims to achieve with its response; 2) Response Generation: Based on the defined intention and dialogue history, each LLM generates an original response; and 3) Alternative Response Generation: Three distinct alternative replies are  produced for each original reply, representing diverse negotiation strategies or perspectives while preserving the original intention. All replies—original and alternatives will be scored on clarity, relevance, engagement, and alignment with the stated intention of the whole dialogue when the negotiation is finished. Dynamic regret analysis was conducted by comparing the scores of the original replies against the highest-scoring alternative responses, providing a *quantitative measure of suboptimality*. This approach allowed for the evaluation of regret dynamics across turns and offered insights into how regret analysis can inform improved decision-making strategies in this negotiation context. We have a detailed example of negotiation in Section C.11.2.
>
> The results are presented in Figure C.11. Our results  indicate that LLMs are able to achieve sublinear dynamic regret according to our trend-checking framework and regression-based framework, in this case study. We have added our paper discussing this case study in Section C.11.2.

---

> ### Author Response · Authors · 2024-11-21
> **Rebuttal by Authors (3)**
>
> ---
>
> ## How sensitive are the findings to the choice of synthetic games
>
> We would like to kindly remind the reviewer that the reason why we use the win-win, prisoner’s dilemma, unfair, cyclic, biased, and second best is that they can represent all possible two-action games (see [8, 15]). Meanwhile, we would like to emphasize that we also have added *randomization* to each type of payoff matrix. Therefore, we believe such a categorization and the corresponding results are representative enough. Finally, we would like to remind the reviewer that, we also did have results for *randomly-generated* games beyond these ones.
>
> ---
>
> ## How do the authors validate the importance of the research problem
>
> To summarize and restate all of our previous responses:
>
> - **Why studying online and strategic decision-making behaviors of LLMs?** We believe this is a research question that has received increasing attention as validated in the following two major lines of literature: (1). The first line of literature studies how to use LLMs for *real-world decision-making tasks*, which requires online and strategic decision-making due to the non-stationary or game-theoretical environments. **And our work can be viewed as grounding such (already delivered successes) with more principles and rigor**. (2). The second line of literature aims to reveal the principle and general law of LLMs in decision-making problems through controlled synthetic experiments in idealized environments. Those studies have spanned from bandit problems to repeated games, which are exactly the settings that necessitate online and strategic decision-making behaviors, and the settings we focused on. Both lines of works justified the significance of the research problem we aimed to tackle.
> - **Why using regret?** With the rich literature that has studied various aspects of LLMs in online learning or game-theoretical tasks in mind, to the best of our knowledge, there are no principled and unified metrics better than regret for both online and game-theoretical problems.  For example, (1). In stationary environments like single-agent RL, regret is equivalent to accumulative rewards as we mentioned before; (2). In non-stationary and potentially adversarial environments, regret is arguably the most standard and fundamental notion [12] and a strong indication of robustness and rational behaviors; (3). In repeated games, regret can guide the independent learning process of the agents, where no-regret behaviors will lead to equilibria (Remark D.1), and are usually viewed as rational strategic behaviors [13,16,17].
>
>
> Again, we  really appreciate the feedback from Reviewer KLuz, and would be more than happy to address any further questions the reviewer may have.

---

> ### Author Response · Authors · 2024-11-21
> **Rebuttal by Authors (4)**
>
> ---
>
> [1] Yubin Kim, et al. "MDAgents: An Adaptive Collaboration of LLMs for Medical Decision-Making." The Thirty-eighth Annual Conference on Neural Information Processing Systems.
>
> [2] Han Ding, et al. "Large Language Model Agent in Financial Trading: A Survey." arXiv preprint arXiv:2408.06361 (2024).
>
> [3] Yancheng Wang, et al. "Recmind: Large language model powered agent for recommendation." arXiv preprint arXiv:2308.14296 (2023).
>
> [4] Likang Wu, et al. "A survey on large language models for recommendation (2023)." URL: https://arxiv. org/abs/2305.19860 (2023).
>
> [5] Akshay Krishnamurthy, et al. "Can large language models explore in-context?." The Thirty-eighth Annual Conference on Neural Information Processing Systems (2024).
>
> [6] Yue Wu, et al. "SmartPlay: A Benchmark for LLMs as Intelligent Agents." The Twelfth International Conference on Learning Representations (2024).
>
> [7] Fanzeng Xia, et al. "Beyond Numeric Awards: In-Context Dueling Bandits with LLM Agents." arXiv preprint arXiv:2407.01887 (2024).
>
> [8] Elif Akata, et al. "Playing repeated games with large language models." arXiv preprint arXiv:2305.16867 (2023).
>
> [9] Jillian Ross, Yoon Kim, and Andrew Lo. "LLM economicus? Mapping the Behavioral Biases of LLMs via Utility Theory." First Conference on Language Modeling.
>
> [10] Mohammad Gheshlaghi Azar, Ian Osband, and Rémi Munos. "Minimax regret bounds for reinforcement learning." International conference on machine learning. PMLR, 2017.
>
> [11] Chi Jin, et al. "Is Q-learning provably efficient?." Advances in neural information processing systems 31 (2018).
>
> [12] Elad Hazan. "Introduction to online convex optimization." Foundations and Trends® in Optimization 2.3-4 (2016): 157-325.
>
> [13] Nicolo Cesa-Bianchi, and Gábor Lugosi. Prediction, learning, and games. Cambridge university press, (2006).
>
> [14] Davide Marchiori, and Massimo Warglien. "Predicting human interactive learning by regret-driven neural networks." Science 319.5866 (2008): 1111-1113.
>
> [15] David Robinson, and David Goforth. The topology of the 2x2 games: a new periodic table. Vol. 3. Psychology Press, (2005).
>
> [16] Avrim Blum, MohammadTaghi Hajiaghayi, Katrina Ligett, and Aaron Roth. Regret minimization
> and the price of total anarchy. In Proceedings of the fortieth annual ACM symposium on Theory
> of computing, pp. 373–382, (2008).
>
> [17] Tim Roughgarden. Intrinsic robustness of the price of anarchy. Journal of the ACM (JACM), 62(5): 1–42, (2015).
>
> [18] Junling Liu, et al. "Is ChatGPT a good recommender? a preliminary study." arXiv preprint arXiv:2304.10149 (2023).
>
> [19] Sahar Abdelnabi, Amr Gomaa, Sarath Sivaprasad, Lea Schönherr, and Mario Fritz. "LLM-Deliberation: Evaluating LLMs with Interactive Multi-Agent Negotiation Games." (2023).

---

> ### Comment · Reviewer_KLuz · 2024-11-22
>
> I appreciate the authors' detailed and thoughtful responses to my questions.
>
> The revisions have addressed my concerns comprehensively, and I am satisfied with the improvements made to the manuscript.

---

### Official Review · Reviewer_Q6rC · 2024-10-28

**Soundness:** 4
**Presentation:** 2
**Contribution:** 3
**Rating:** 6
**Confidence:** 3

**Summary:**

The paper discusses the use of LLMs for decision-making in both single and multi-agent settings. Their study focuses on understanding the limitations of LLM agents in interactive environments by using regret as a performance metric. Initial empirical investigations assess no-regret behaviors in non-stochastic online learning and the emergence of equilibria in repeated game scenarios. The paper provides theoretical insights into these behaviors based on assumptions about the data’s origin and the rationality of human decision-makers involved in pre-training.

**Strengths:**

1. The paper explores an interesting aspect of using LLM in decision-making tasks. Experimental results are provided.

2. It is interesting that Theorem 5.2 shows a single-layer (linear) self-attention transformer is enough to model no-regret learning algorithms.

3. The paper provides an extensive appendix. The appendix is rich with additional proofs, experimental details, and supplementary discussions. I didn't read all the experimental results and proofs.

**Weaknesses:**

1. It seems not that natural to apply LLMs to solve online decision-making problems, given the existence of classic and well-established algorithms. What are the potential applications? The paper needs to be better motivated to help readers understand the contributions. While the application of LLMs in online decision-making presents intriguing possibilities, the practical implications are not immediately clear. To better engage readers and clarify the significance of this research, the paper could benefit from a more compelling motivation. Specifically, detailing potential real-world applications and explicitly connecting these to the experimental findings would enhance the paper’s relevance and impact.

It's worth considering providing specific examples of potential real-world applications where using LLMs for online decision-making could offer advantages over classical algorithms such as FTPL.

2. The paper does not address the capabilities of state-of-the-art LLM APIs like GPT4 Turbo and o1. Discussing their performance might also offer valuable insights into the scalability and applicability of the proposed methods to more complex decision-making tasks. More detailed analysis of this discrepancy are needed.



Overall, I'd like to raise my score if case studies on real-world application can be provided, with both GPT-4 Turbo and o1 evaluated.

**Questions:**

1. When using GPT4 Turbo and GPT o1 for online decision-making, could the assumption on optimal action labels be relaxed?

2. Why is the regret performance of GPT4 better than single/multi-layer models in the single agent setting, yet vice versa in the multi-agent setting? What factors contribute to this discrepancy in its effectiveness across different settings?

3. What is the technical challenge in the proof of Theorem 5.2? If the result goes beyond simply applying standard learning theory arguments, it's worth highlighting the proof techniques used or created.

---

> ### Author Response · Authors · 2024-11-21
> **Rebuttal by Authors (1)**
>
> We thank Reviewer Q6rC for the detailed comments and insightful feedback. We address Reviewer Q6rC's concerns and questions below:
>
> ## Regarding the significance of understanding LLMs in online learning tasks
>
> We believe our studies are significant in the following aspects.
>
> - **From the perspective of *understanding empirical LLM-agent applications*:**
> There has been a large and fast-growing body of literature on *LLM agents* for decision-making, where LLMs are employed as an online decision-maker in various real-world applications. This includes medical decision-making [1], financial trading [2], recommendation systems [3, 4], etc. In all these  domains, promising successes have been shown in  environments that are sequential, non-stationary, and strategic, which is exactly the setup formally investigated in Online Learning and Game Theory. **Therefore, we would like to emphasize that our goal was not to be the first to *propose using LLMs for online decision making* problems, but to systematically *understand* some already-promising successes of LLMs in online decision-making, through canonical and controlled experiments, under a unified, principled framework, and with some theoretical insights.** It is worth emphasizing that this kind of research efforts have been also deemed valuable as we will explain in the next bullet point.
> - **From the perspective of *principled understanding of LLMs' capabilities/fundamental limits*:**
> Understanding the online decision making ability of LLMs through **numerical/synthetic problems in a more principled manner** have also received increasing attention. To name a few, [5, 6] studies LLMs for multi-arm bandits, [7] studied LLMs for dueling bandits, and [8, 9] studied LLMs in game-theoretical tasks. **All of such works including ours primarily create synthetic data, build an idealized environment, and perform a large number of controlled experiments, aiming to provide deeper understandings of LLMs' capabilities and limits, instead of focusing on proposing new methods/improving existing approaches for certain real-world tasks.** Note that none of any those references include any experiments on large-scale, real-world applications, because of such a different focus. Such a philosophy is in fact also well-recognized as studying the physics of LLMs [10].
>
> - **From the perspective *Economic and Simulation studies*:**
> Apart from all the applications that the powerful tool of Online Learning and Game Theory can solve, our paper also provides valuable insights into  economic studies in this era of LLMs. Specifically, economists have been using LLMs to replace humans to perform economic studies [9, 11], considering the high costs of collecting human data. Note that such economical studies/surveys also only involve simple synthetic problems, but are still deemed valuable, significant, and instructive. However, the rationality of LLMs and its resemblance to humans has been less studied. Our paper is exactly evaluating the rationality of LLMs through the lens of *regret* (which is also a widely-used notion in economics and game theory), and revealing its resemblance to humans by the quantal response model (in Sec. 4).
> - **From the perspective of *potential benefits of LLMs over classical algorithms*:**
> Finally, **even in terms of solving canonical Online Learning and Games problems, our studies may provide some new insights into the potential benefits of LLMs over classical no-regret algorithms**, as a result of our controlled experiments. Specifically, classical algorithms like FTRL/FTPL can better handle the problems in the extreme worst cases, while LLMs **can outperform such algorithms when the loss sequences exhibit certain trends** (see our Sec. 3.4 and Appendix C.7). This is due to the **in-context learning ability** of LLMs (see a detailed discussion in our response to the last question of Reviewer d5nu), for which classical algorithms do not possess. This highlights one notable benefit of employing LLMs as decision-makers instead of classical algorithms,  since many real-world applications are not necessarily in the  worst-case.
> ---

---

> ### Author Response · Authors · 2024-11-21
> **Rebuttal by Authors (2)**
>
> ## Regarding scalability and applicability of the proposed methods to more complex decision-making tasks
>
>
> - Firstly, we would like to point out that **we have studied GPT-4-Turbo** in Sec. 3.4, and revealed that even such a stronger model can still fail in the worst-case adversarial environments we constructed. This motivated us to further study how to further **enhance**  the online learning ability of transformer models later in Sec. 5.
> - More importantly, we want to kindly remind the reviewer that we are not ***proposing a new way of utilizing LLMs***, but rigorously evaluating/understanding the ***native*** ability of LLMs in Online Learning and Game Theory problems (along with some standard intervention techniques like CoT prompting). As in our response to the first question, we, as well as a bunch of related studies [5-9], are trying to understand the **fundamental question** of why LLMs can perform relatively well in challenging domains that requires sequential, non-stationary, and strategic decision making, **through controlled experiments in canonical  environments, so that we can do some quantitative analyses**. In other words, the essence of ours and [5-9] is to abstract the problems first and discover the basic behavioral patterns of LLMs, while scalability and applicability are the focus of papers like [1-4], *which targeted certain applications and are thus perpendicular to our focus*. After all, we did **not** claim to have ***proposed new methods to solve certain real-world problems*** in the paper, as it is not the focus of this type of research (and has been done very well in works like [1-4]).
>
> ---
>
> ## Real-world case studies using GPT4-Turbo and o1 (added in Appendix C.11 of our revised paper)
> As what we kindly reminded the reviewer before, we are **not** proposing a **new method to utilize LLMs** to solve **certain application tasks**, but rather to **evaluate** the rationality and **understand** the success of LLMs in sequential, non-stationary decision-making environments through **regret**. That being said, as per your suggestion, **we have managed to also obtain some results for real-world case studies, using more SOTA models of GPT4-Turbo and o1** (see below). Given the focus of our paper, a meaningful evaluation on real-world case studies that could strengthen our message would be to validate whether our findings that LLMs can be no-regret still hold in some real-world case studies (although existing related works [5-8] that shared the same research focus have **only** done evaluation on **synthetic**  problems).
>
> ### Case Study 1) Sequential Recommendation.
>
> Towards such a goal, we consider the task of **sequential recommendation**, a task that people have been employing LLMs to solve with success. Note that how existing literature (e.g. [3, 4, 12]) uses LLMs to solve this task **exactly fits our Online Learning framework**, where humans feed a *history of items the user has interacted with* to the LLM, and then ask the LLM to *recommend the item* (or several items) the user may want to interact next, and the entire process carries on repeatedly. We refer to [12] for a more detailed introduction. We use the real-world data and follow the experimental setup of [12], and  evaluate the regret of such an online learning process. We refer the detailed introduction to the new experiments and results to Appendix C.11 of our revised paper.
>
> From the results, we can see that LLMs are indeed no-regret **on such a real-world application with real-world data**. As a comparison, we also replace the real-world data with synthetic data generated in a uniformly random way, where  similar no-regret behaviors can still be observed. Also, interestingly, **a new insight that has come up is that LLMs even perform better on real-world data, which validates that real-world data may exhibit certain trend/structures, for which LLMs can exploit and thus, achieve superior performance, as we also showed in our paper.**

---

> ### Author Response · Authors · 2024-11-21
> **Rebuttal by Authors (3)**
>
> ### Case Study 2) Interactive Negotiation.
>
>
> The experiment was designed to simulate negotiation scenarios between two agents, designated as LLM A and LLM B, over multiple turns. While previous studies have explored negotiation using LLMs (e.g.,  [13]), our setup differs by focusing on *language-based*   negotiation problems rather than *numerical* negotiation problems, which is more natural and closer to real-world applications. This shift allows for a more diverse exploration of negotiation scenarios, moving beyond purely quantitative frameworks.
>
> In each turn, a unique negotiation topic was generated using LLMs, providing context, objectives, and relevant background information. The negotiation is executed in a turn-based manner, where each turn consists of three steps: 1) Intention Generation: Each LLM defines its goal for the turn, specifying what it aims to achieve with its response; 2) Response Generation: Based on the defined intention and dialogue history, each LLM generates an original response; and 3) Alternative Response Generation: Three distinct alternative replies are  produced for each original reply, representing diverse negotiation strategies or perspectives while preserving the original intention. All replies—original and alternatives will be scored on clarity, relevance, engagement, and alignment with the stated intention of the whole dialogue when the negotiation is finished. Dynamic regret analysis was conducted by comparing the scores of the original replies against the highest-scoring alternative responses, providing a *quantitative measure of suboptimality*. This approach allowed for the evaluation of regret dynamics across turns and offered insights into how regret analysis can inform improved decision-making strategies in this negotiation context. We have a detailed example of negotiation in Section C.11.2.
>
> The results are presented in Figure C.11. Our results  indicate that recent GPT models (e.g., GPT-4 Turbo, o1) are able to achieve sublinear dynamic regret according to our trend-checking framework and regression-based framework, in this case study. We have added our paper discussing this case study in Section C.11.2.
>
>
> ## Could the assumption on optimal action labels be relaxed?
>
> - Firstly, we believe the actual training process of stronger models like GPT4 Turbo and GPT o1 often involves collecting data of *higher quality*. Therefore, the assumption on optimal action labels might only be easier to satisfy in general.
> - Secondly, the assumption we propose is just *one hypothetical explanation* for why even relatively *weaker* LLMs like GPT-4 and GPT-3.5-Turbo, although not trained for online decision-making, have exhibited no-regret behaviors on a large portion of Online Learning problems we tested. The behavior predicted by the theorem under this assumption is also validated in Figure 4.1. Therefore, we do not see too much necessity to relax this assumption for stronger models trained by datasets of higher quality, under our current theoretical framework.
> - Towards relaxing this assumption (not only for GPT-4-Turbo and o1), we have also proposed an alternative explanation for LLMs on problems *with certain trends*. This alternative explanation is based on the assumption that LLMs have adopted the in-context learning capability  (see our detailed discussion in the response to the last question of Reviewer d5nu), which may be a potential angle to relax such an assumption. But again, as we mentioned in that response, such trend-predicting behaviors are not generally observed, and the associated explanation is not generally applicable for loss sequences *without any trend*  (unlike our explanation based on the connection to no-regret learning algorithms).

---

> ### Author Response · Authors · 2024-11-21
> **Rebuttal by Authors (4)**
>
> ---
>
> ## Why is the regret performance of GPT4 better than single/multi-layer models in the single agent setting, yet vice versa in the multi-agent setting? What factors contribute to this discrepancy in its effectiveness across different settings?
>
> As the reviewer pointed out, LLMs can sometimes outperform FTRL/FTPL algorithms and single/multi-layer models. This phenomenon primarily occurs when the loss sequence follows a certain trend, as seen in the single-agent setting. In Section 3.4, we provided explanations for this behavior through canonical counterexamples for the follow-the-leader algorithm. Specifically, we highlighted that when the loss sequences exhibit *obvious or predictable patterns*, LLMs can accurately infer the next loss vector based on the historical data, enabling them to make *near-optimal*  decisions. We have further formalized this discussion using the perspective of in-context learning (see the response to the last question of Reviewer d5nu). In contrast, FTRL/FTPL, due to their update rules, tend to output near-uniform policies in these cases, so do  single/multi-layer Transformer models.
>
> Additionally, in Appendix C.7 of our submission, we have conducted ablation studies to validate that LLMs leverage trends in these examples by comparing their performance when provided with *raw history* versus *summarized history* (i.e., the summation of historical loss sequences). Notably, after such a summarization, the summarized loss vector *no longer reflects the trend*, **leading to significantly worse** performance of the LLMs. We have further clarified and emphasized these findings in the updated manuscript.
>
> In stark contrast, in the game or multi-agent setting, the loss sequence trend **depends on the behavior of other agents**, making it inherently more unpredictable as all agents continuously update their behavior policies. This unpredictability likely explains why LLMs perform no better -- or only comparably -- to FTRL/FTPL algorithms or single/multi-agent-trained transformers in such settings. We appreciate your careful reading of our results and the insightful question, and have added such a discussion in the updated paper Section E.11.
>
>
> ## What is the technical challenge in the proof of Theorem 5.2? If the result goes beyond simply applying standard learning theory arguments, it's worth highlighting the proof techniques used or created.
>
> First, Theorem 5.2 addressed  the **optimization guarantees**,  while Theorem 5.1 is the one that focused more on the **(statistical) learning theoretic guarantees**. We summarize the technical novelties for both results below.
>
> - **Theorem 5.1**: Since we are using a surrogate loss function, it was necessary to establish the *uniform convergence* of the regret loss function. To address this, we introduced several novel techniques, including Lemma 8 and Lemma 9, which employed truncation to handle the uniform convergence of the regret loss function. These techniques involve a completely new approach, particularly for handling the *double-iterated limit*. Moreover, we constructed the constrained optimization problem as stated in Claim 5,  and employed **Berge's Maximum Theorem** -- an atypical method -- to establish the Lipschitz continuity of the loss functions.
>
> - **Theorem 5.2**: This theorem involves the analysis of a **non-convex optimization**  problem through stationary point analysis. We identified the set of stationary points through a step-by-step process (Steps 1, 2, and 3 in Section E.7). By constructing the optimization problem as shown in Equation E.13, we significantly *reduced the candidate set* for optimal points using our novel argument on the expected value of a nonnegative definite matrix (see page 68-69).  The main challenge here was to address the global optimization problem in a non-convex setting, which required some good understanding of the  Transformer architecture.
>
> We have highlighted these challenges and contributions with more details in our revised manuscript Section E.7.
>
> ---
>
> Again, we  really appreciate the feedback from Reviewer Q6rC, and would be more than happy to address any further questions the reviewer may have.

---

> ### Author Response · Authors · 2024-11-21
> **Rebuttal by Authors (5)**
>
> ---
>
> [1] Yubin Kim, et al. "MDAgents: An Adaptive Collaboration of LLMs for Medical Decision-Making." The Thirty-eighth Annual Conference on Neural Information Processing Systems.
>
> [2] Han Ding, et al. "Large Language Model Agent in Financial Trading: A Survey." arXiv preprint arXiv:2408.06361 (2024).
>
> [3] Yancheng Wang, et al. "Recmind: Large language model powered agent for recommendation." arXiv preprint arXiv:2308.14296 (2023).
>
> [4] Likang Wu, et al. "A survey on large language models for recommendation (2023)." URL: https://arxiv.org/abs/2305.19860 (2023).
>
> [5] Akshay Krishnamurthy, et al. "Can large language models explore in-context?." The Thirty-eighth Annual Conference on Neural Information Processing Systems (2024).
>
> [6] Yue Wu, et al. "SmartPlay: A Benchmark for LLMs as Intelligent Agents." The Twelfth International Conference on Learning Representations.
>
> [7] Fanzeng Xia, et al. "Beyond Numeric Awards: In-Context Dueling Bandits with LLM Agents." arXiv preprint arXiv:2407.01887 (2024).
>
> [8] Elif Akata, et al. "Playing repeated games with large language models." arXiv preprint arXiv:2305.16867 (2023).
>
> [9] Jillian Ross, Yoon Kim, and Andrew Lo. "LLM economicus? Mapping the Behavioral Biases of LLMs via Utility Theory." First Conference on Language Modeling.
>
> [10] https://physics.allen-zhu.com/
>
> [11] John J. Horton,  Large language models as simulated economic agents: What can we learn from homo silicus?. No. w31122. National Bureau of Economic Research, 2023.
>
> [12] Junling Liu, et al. "Is ChatGPT a good recommender? a preliminary study." arXiv preprint arXiv:2304.10149 (2023).
>
> [13] Sahar Abdelnabi, Amr Gomaa, Sarath Sivaprasad, Lea Schönherr, and Mario Fritz. "LLM-Deliberation: Evaluating LLMs with Interactive Multi-Agent Negotiation Games." (2023).

---

> ### Author Response · Authors · 2024-11-24
> **Comment**
>
> We sincerely appreciate the valuable feedback you provided. We wanted to follow up to ensure that our revisions have adequately addressed your concerns. If you have any additional comments or questions, we would be grateful if you could share them with us.
>
> Thanks for your time and considerations!
>
> Best regards,
>
> The Authors

---

> > ### Comment · Reviewer_Q6rC · 2024-11-25
> >
> > Thank you for answering my questions. Based on the revised experiments, I'd like to raise my score to 6.

---

### Official Review · Reviewer_d5nu · 2024-11-03

**Soundness:** 4
**Presentation:** 3
**Contribution:** 3
**Rating:** 8
**Confidence:** 4

**Summary:**

The authors investigate whether LLMs can exhibit no-regret behavior in online learning scenarios. Their main contributions include:

- **Empirical Validation:** The authors empirically show that both commercial and open-source LLMs can achieve no-regret learning in various environments, including dynamic settings and multi-agent games. They demonstrate that LLMs often outperform traditional no-regret algorithms like FTRL and FTPL, with the models effectively incorporating randomization in decision-making.
- **Limitations in Adversarial Scenarios:** The study identifies cases where advanced LLMs, such as GPT models, struggle to achieve sublinear regret, particularly in highly adaptive or adversarial environments.
- **Theoretical Insights and Enhancements:** The authors provide a theoretical framework explaining how next-token prediction during pretraining enables LLMs to develop capabilities aligned with no-regret algorithms. They also introduce a novel "regret-loss" training objective designed to further strengthen the no-regret properties of LLMs, even in the absence of optimal action labels.

**Strengths:**

The strengths of the paper include:

- **Comprehensive Empirical Analysis:** The authors conduct a thorough empirical study across diverse environment setups and various LLM models. They evaluate LLM performance in both standard online learning scenarios and multi-agent games, identifying cases where LLMs exhibit no-regret behaviors and outperform classical algorithms like FTRL and FTPL. Importantly, the authors also examine situations where LLMs, such as GPT, fail to achieve sublinear regret in highly adaptive or adversarial environments. This balanced examination provides valuable insights into the conditions under which LLMs succeed or struggle, helping us better understand the decision-making processes and strategic adaptations LLMs employ in online game contexts.

- **Conduct Ablation Study:** The authors conduct an ablation study to systematically examine the contributions of various components in their proposed framework.

- **Theoretical Justification for No-Regret Behaviors:** The authors offer a theoretical explanation on how next-token prediction pretraining enables LLMs to develop no-regret behaviors in online learning. Under certain structured noise assumption, the optimal behaviors of the LLM under the next-token prediction loss can be related to FTPL.

- The no-regret loss seems to be novel, and enhances the model's performance.

**Weaknesses:**

I found the following weakness points:

- **Ambiguity in No-Regret Definition and Experimental Setup:** The standard definition of "no-regret" in online learning typically requires that an algorithm achieves sublinear regret for all possible loss sequences, including adversarial ones. However, the authors claim that the LLM behaves as a no-regret algorithm based primarily on experiments involving uniform or Gaussian losses and losses with specific trends, which are not adversarial. This raises some confusion, as these setups do not fully reflect the robustness expected in no-regret algorithms. In fact, the experiments on the "Adaptive loss sequence" seem more directly relevant to evaluating this claim.

- **Lack of Explanation for Model Performance:** Following the previous point, while the authors show that the LLM outperforms traditional no-regret algorithms like FTRL and FTPL, there is limited insight into why this occurs. For instance, the model may be learning the underlying distribution of the loss or fitting specific loss trends, but this is not explicitly demonstrated. Additional analysis on how GPT models achieve superior performance would provide a more comprehensive understanding of these results.

- **Theoretical Limitation**: In Theorem 5.2, you only show the results for $N=1$. If my understanding is correct, this basically means that there is no maximum taken across the input loss sequence $\ell_{1:t}$. Is this also the setup for training the single/multi-layer transformer model under the noisy alternating/adaptive loss as you show in Figure 3.4, or did you use larger $N$ to produce the results? I didn't find the experiment setup for these results in Figure 3.4.

- Building on the previous point, there is one minor thing. I feel that Theorem 4.1 may not fully capture the behavior of LLMs. The empirical results show that GPT models outperform standard no-regret algorithms in more structured tasks but perform significantly worse in adversarial settings. However, what this theorem suggests is that the model after pretraining performs something similar to FTPL. This discrepancy suggests that the theoretical framework might not align with the model's observed strengths and weaknesses across different environments.

However, I still find the contributions to be significant. Hence, I'm open to raising my scores upon the concerns being addressed.

**Questions:**

1. Could the authors clarify the use of the term "no-regret" in this context?
2. In structured, non-adversarial environments (such as linear or sine-trend losses), GPT-4 appears to outperform traditional no-regret algorithms like FTRL/FTPL. Does this suggest that the model may be doing more than no-regret learning, possibly adapting to or fitting the specific trends in the loss sequences through in-context learning?

**Details Of Ethics Concerns:**

No concerns.

---

> ### Author Response · Authors · 2024-11-21
> **Rebuttal by Authors (1)**
>
> We thank Reviewer d5nu for the detailed comments and insightful feedback. We are encouraged that the reviewer finds "empirical study to be thorough" and "the no-regret loss to be novel". We address Reviewer d5nu's concerns and questions below:
>
> ## Ambiguity in no-regret definition and experimental setup
>
> - **Regarding the terminology of no-regret:** We thank the reviewer for pointing out the potentially inconsistent use of the terminology *no-regret* from the literature. We did understand that no-regret algorithms should ensure the regret to grow sublinearly for arbitrary $\ell_{1:T}$, while we have indeed found certain loss sequences, under which LLM's regret grows linearly (see e.g., Sec. 3.4 where we specifically focused on this aspect). Indeed, it might be difficult to expect LLMs to achieve sublinear regret in *any* loss sequences, given they were such complicated systems and were *not trained* for it. Therefore, when we use the terminology of *no-regret behaviors* for certain experiments/plots, we actually meant that *the regret grows sublinearly* (under the presented loss sequences.) In fact, in our initial introduction of being no-regret in Sec. 2.2, we introduced it as a property for a loss sequence $(f_t)\_{t\in[T]}$(, where now we have changed the definition of no-regret to be for all possible $(f_t)_{t\in[T]}$ in the revision). We really appreciate your feedback on this point, and have clarified this point and made changes accordingly (from *no-regret* to *regret grows sublinearly* (for the loss sequences considered)) overall in the updated manuscript.
> - **Clarifications for uniform/gaussian random loss sequences:** We remark that the reason why we have considered loss sequences generated from uniform/gaussian distributions is as follows:
>     - We use them as a concrete and actionable  way of searching over all possible inputs since both distributions have full support on the problem space, and can be generated conveniently. Note that, the regret notion we cared about (standard in this Online Learning setting) is  **for each realization** of the loss sequence, and the statistical model that generated them is *not relevant*. Achieving sublinear regret on those randomly generated (but *arbitrarily different* over time) loss sequences also serve as a good indicator that the LLMs can achieve sublinear regret for *a large portion* of the problems in the entire problem space, which is still meaningful for examining/understanding the online decision-making properties of LLMs, as most real-world examples may **not be that adversarial** (while the regret-notion and sublinear-regret behavior are **still well-defined and meaningful**).
> Finally, as mentioned above, we never expected  (pre-trained, off-the-shelf) LLMs to achieve sublinear regret for **any** loss sequences, either, which was exactly why we had Sec. 3.4 with counterexamples, and Sec. 5 with new Regret-Losses to further promote the no-regret behavior.
>     - We emphasize that the (realized) loss sequences generated in such a way do not exhibit certain obvious patterns (we believe it is difficult for either LLMs or humans to observe that the loss sequences are drawn from a uniform/gaussian distribution). Therefore, it can better help us inspect and conclude the *general behavioral pattern* of LLMs, e.g., computing the sum/average of the loss sequences and taking randomized policies, which actually inspires us to build the quantal response model to model the behavior of LLMs.

---

> ### Author Response · Authors · 2024-11-21
> **Rebuttal by Authors (2)**
>
> - **Motivations for studying various loss sequences beyond the worst cases:**
>     - As mentioned in our first point, studying *worse-case* no-regret guarantee is certainly one important aspect for online learning. However, LLMs as a rather complex and random system can make occasional errors due to various kinds of reasons like hallucination, arithmetic errors, misunderstanding of the tasks in the natural language form, which makes it not very viable (nor interesting) to *always achieve the worst-case* no-regret guarantee *only through experiments* (as they may always fail in the worst case, as our Sec. 3.4 has examplified). Therefore, to help us understand the worst-case regret guarantee of LLMs in a more reasonable way, we decide to build an abstract mathematical model that can capture the LLM's behaviors relatively accurately, and study the theoretical guarantees of the abstract model instead of the raw LLMs. This model together with our empirical observations may provide some insights into LLMs' online learning capabilities, e.g., computing the loss summation and being randomized (see also the fitted behaviors in Fig 4.1).
>     - Secondly, studying regret beyond the worst-cases is still valuable in both theory and practice for Online Learning. For example, in theory, [1, 2] have studied the online learning problem in the smoothed analysis setting under various assumptions on the adversaries. More importantly, in practice, the adversaries are oftentimes not the worst-case for various reasons, e.g. physical constraints, limited computational power for adversaries, or environment being just random instead of adversarial.
>     - More importantly, for another half of our motivation -- *playing repeated games in multi-agent settings*, the opponent players who follow some algorithms for their own decision-making are usually not adversarial. In fact, such a "less-adversarial" property is well-acknowledged in game-playing for achieving better *no-regret* guarantees, see e.g., [4, 5]. Therefore, we believe that investigating the performance under loss sequences that are not necessarily worst-case is still valuable, especially given that our work is an initial attempt towards carefully understanding the no-regret behaviors of LLMs in Online Learning and Games.
>
> - **Regarding the claim that LLM behaves as a no-regret algorithm:** As in the previous bullet point, we would like to clarify that our claim that LLMs behaves like the no-regret algorithms are not purely based on the experiments on uniform/gaussian (and games), but also the behavioral pattern we abstract from experimental data (line 435-441)
>
> ---
>
> ## Lack of Explanation for Model Performance
>
> As what the reviewer has pointed out,  LLMs can sometimes outperform the FTRL/FTPL algorithms. This kind of phenomenon mainly happens when the loss sequence follows a certain trend. In fact, we believe we have provided related explanations through the canonical counterexamples for follow-the-leader in Sec 3.4. Specifically, we have pointed out that when the loss sequences exhibit certain *obvious/predictable patterns*, LLM can accurately identify the next loss vector it will receive according to the history, and thus can easily make near-optimal decisions. We shall discuss it later more formally using the perspective of in-context learning. In contrast, FTRL/FTPL according to their update rules will only output near-uniform policies.
>
> Furthermore, in Appendix C.7, we also performed ablation studies to validate that the LLMs are really harnessing the trend in those examples, by comparing the performance of LLMs when providing the *raw history* versus providing only the *summarized history* (i.e., the summation of history loss sequences). Note that after such a summarization, such a single summarized loss vector cannot reflect the *trend* anymore and the LLM performs much worse correspondingly. We have further clarified and emphasized these findings and explanations in the updated manuscript Section E.11.
>
> ---

---

> ### Author Response · Authors · 2024-11-21
> **Rebuttal by Authors (3)**
>
> ---
>
> ## Regarding Theoretical Limitation
> We established the regret bound for general $N$ in Theorem 5.1. Theorem 5.2 focused on the *single-layer* Transformer  case, in order to provide some **optimization guarantees** of our regret loss, drawing a parallel to the **statistical guarantees** in Theorem 5.1. To analyze the *optimization* property of our regret-loss, we do need some particular form of the loss with some particular $N$, which was why the analysis in Theorem 5.2 focused on one such case of $N = 1$. The parameter $N$ is associated with the *surrogate loss function*, which *approximates the maximum regret* value with respect to $\ell_{1:t}$. As outlined in our paper, directly differentiating the maximum regret value is computationally challenging, prompting us to adopt this alternative and novel formulation, with provable guarantees. Notably, we still provide a theoretical guarantee for optimizing the maximum regret value through the surrogate loss function.
>
> Furthermore, we presented the results of training **both** single-layer and multi-layer Transformers with our regret-loss for $N = 2, 4$ and $k = 1, 2$ in Figures E.6, E.7, and E.8. These results demonstrated comparable or, in some cases, even *superior* regret performance relative to GPT-4 or established no-regret learning algorithms. Figure 3.4 specifically illustrates the results for $N = 1$; we have added clarification for  this distinction in our revised Figure.
>
> ---
>
> ## Regarding whether Theorem 4.1 can fully capture the behavior of LLMs
>
> This is in fact related to our response to the second weakness. We firstly remark that LLMs can exhibit very diverse behaviors, where for example, the trend-predicting behavior mentioned in our response to the second weakness is one behavior pattern not captured by our Theorem 4.1. Such a trend-predicting behavior is indeed a major strength of LLMs compared with FTRL/FTPL (for particular loss sequences, not generally applicable/triggered), which we have further emphasized in our revision (Section E.11). Meanwhile, we believe there could also be other possible (but probably rare) patterns not identified yet.
>
> However, the reason why we did not emphasize or theoretically study other explanations of LLMs' good/no-regret  behaviors is that, they may be relatively **less common and more fragile, and not as generally applicable as that we studied in Theorem 4.1**. For example, when we add some noises to problems with certain trends, i.e., the noisy alternating loss sequence we have proposed in Sec. 3.4, the performance of LLMs becomes significantly worse. In contrast, the behavior our Theorem 4.1 predicted can accurately predict the behaviors of LLMs for the large family of problems in Sec. 3.2, see e.g., Figure 4.1.
>
> Finally, we admit that although the behavior identified in Theorem 4.1 is much more generally applicable to model LLMs than the *trend-predicting* one, such a behavior is still not necessarily always triggered (even *without* trends in the loss sequences).
>
> Our Theorem 4.1, as we emphasized in the paper, is highly hypothetical and meant for a large portion of (instead of *any possible*) cases, given that LLMs are highly complex and randomized systems, with sometimes unpredictable behaviors. These may exactly be  the reason for the regrettable behaviors we have observed in.

---

> ### Author Response · Authors · 2024-11-21
> **Rebuttal by Authors (4)**
>
> ## Regarding the in-context learning explanation for problems with trends.
>
> We thank the reviewer for bringing up the insightful perspective of in-context learning. We believe this is exactly the explanation for the superior performance of LLMs on loss sequences with trends. Specifically, the task of predicting $\ell_{T+1}$ given a past loss sequence $\ell_{1:T}$ could be exactly equivalent to an in-context learning problem as follows:  The demonstration/in-context dataset is given by the following input and label pairs
> $$
> D=\\{x_t, y_t\\}\_{t\in[T-1]},
> $$ where $x_t=\ell_{1:t}$ and $y_t=\ell_{t+1}$ for each $t\in[T-1]$. Then, LLMs given such demonstration/context $D$ will make prediction based on $x_T=\ell_{1:T}$ (to predict $y_T$, i.e., the next loss vector $\ell_{T+1}$). In other words, in-context learning in this case is exactly firstly learning the trend from the $T-1$ pairs of inputs and labels, i.e., learning the *latent concept* (using the terminology from the in-context learning literature [3]), and then make the prediction. Finally, as we mentioned before, the reason why we haven't emphasized this kind of explanation is since it might be only applicable to problems with trends that are simple and obvious enough for LLMs to identify under limited amount of demonstrations. More importantly, such a trend-predicting process cannot achieve sublinear regret in (the large portion of) cases when there is no such a trend, and thus cannot explain the observed no-regret behaviors observed in these cases. In contrast, our explanation in Theorem 4.1, through a connection to FTPL and quantal-response model, may be *more generally applicable* as an explanation.
>
> ---
>
> Finally, we really appreciate all the insightful and constructive feedback, and especially appreciate your careful   reading and understanding of our work. We hope our explanations have addressed your concerns, and would be more than happy to address any further questions that you may have.
>
> ---
>
> [1] Alexander Rakhlin, Karthik Sridharan, and Ambuj Tewari. "Online learning: Stochastic, constrained, and smoothed adversaries." Advances in neural information processing systems 24 (2011).
>
> [2] Nika Haghtalab, et al. "Oracle-efficient online learning for beyond worst-case adversaries." arXiv preprint arXiv:2202.08549 (2022).
>
> [3] Sang Michael Xie, et al. "An explanation of in-context learning as implicit bayesian inference." arXiv preprint arXiv:2111.02080 (2021).
>
> [4] Constantinos Daskalakis, Maxwell Fishelson, and Noah Golowich, "Near-optimal no-regret learning in general games." Advances in Neural Information Processing Systems 34 (2021).
>
> [5] Ioannis Anagnostides, Constantinos Daskalakis, Gabriele Farina, Maxwell Fishelson, Noah Golowich, and Tuomas Sandholm. "Near-optimal no-regret learning for correlated equilibria in multi-player general-sum games." In Proceedings of the 54th Annual ACM SIGACT Symposium on Theory of Computing, (2022).

---

> > ### Comment · Reviewer_d5nu · 2024-11-22
> >
> > Thank you for your thoughtful responses. My concerns have been largely addressed. However, I believe it would be beneficial to include the aforementioned limitations on the theories and your insights through in-context learning (ICL) in the limitations/discussion section for a more balanced presentation.
> >
> > Additionally, in your response: "Furthermore, in **Appendix C.7**, we also performed ablation studies to validate that the LLMs are really harnessing the trend in those examples, by comparing the performance of LLMs when providing the raw history versus providing only the summarized history," did you perhaps mean Appendix C.8? If so, I would suggest referring readers to the corresponding experiments when presenting the results in Section 3.4, as the ablation study provides additional empirical evidence that complements the discussion.

---

> ### Author Response · Authors · 2024-11-22
> **Thank you.**
>
> We sincerely appreciate your thoughtful suggestion and have revised the paper as per your suggestion. Specifically,
>
> (1). We have included the possible explanations through in-context learning in Appendix C.7 and admitted the possibility of other explanations and limitations of our current Theorem 4.1 in Appendix F, the discussion and limitation section.
>
> (2). We have explicitly referred readers to the experimental results for comparisons between utilizing raw history and summarized history in Sec 3.4 of our main paper
>
> Please let us know if you have any further suggestions or concerns. We would much appreciate it if you may update your evaluation if all your concerns have been addressed. Thank you again for your valuable feedback and in-depth engagement, which has significantly improved our paper.

---

> > ### Comment · Reviewer_d5nu · 2024-11-22
> >
> > I've increased my score accordingly.

---

> > > ### Author Response · Authors · 2024-11-23
> > > **Thank You!**
> > >
> > > Again, we would like to thank the reviewer for all the insightful and valuable feedback.

---

### Official Review · Reviewer_qXHf · 2024-11-04

**Soundness:** 2
**Presentation:** 3
**Contribution:** 3
**Rating:** 6
**Confidence:** 2

**Summary:**

This paper studies large language models from a game-theoretic perspective. Observing that LLM-agents are becoming prevalent, they investigate the performance of such LLM agents in terms of regret.

Empirical findings include:

- LLMs exhibit sublinear regret in certain online learning problems
- equilibria emerge when LLM agents interact through repeated games

The authors also develop theory that is consistent with the empirical observations. Finally, the authors propose a new loss function tailored towards regret minimization. They study models optimized under this loss function both theoretically and empirically.

**Strengths:**

- Given the amount of information conveyed, overall the paper is relatively well written, with a sound structure that helps guiding the reader through the large number of findings.
- The contributions appear to be quite substantial. Not being a domain expert, I cannot make a confident judgment as to originality and significance.
- There is a good combination of theory and experiments. The experiments are comprehensive, and the theory appears to be relevant & meaningfully complement the empirical findings.

**Weaknesses:**

- There is a case to be made that the authors are trying to describe too many findings in the main paper. Some figures are hard to read (Figure 3 is a good example of this). The paper ends rather abruptly, with what is essentially a list of pointers to the supplementary material. It appears to me that the layout is deliberately compressed to gain space (e.g., page 10: space above header of Sec. 5.4, space around Eq. 5.3, math fontsize in Thm 5.1).
    - I am genuinely impressed by how comprehensive the study is (there are 77 pages when including the supplementary material). However, I think the main text would benefit from being simplified by omitting certain results. Remember that this is a conference submission with limited space.
    - Again, _conditioned on the set of results presented in the main text_, I don't see many opportunities to improve the organization and presentation, which I find rather good.
- Proposition 1 is problematic. As far as I understand, this proposition rests on an unwritten assumption that the indicators for _difference of regrets is > 0_ are independent. I don't see why this would hold?

**Questions:**

The authors study instruction-tuned models. These typically undergo multiple training stages: a pre-training stage consisting of next-token prediction, and multiple fine-tuning stages that align models with desired behavior.

Have the authors considered comparing the behavior of LLMs at various stages of the training process? This question strikes me as relevant in the context of the discussion in Section 4.2, which makes assumption on the pre-training stage specifically.

---

> ### Author Response · Authors · 2024-11-21
> **Rebuttal by Authors (1)**
>
> We thank Reviewer qXHf for the detailed comments and insightful feedback. We are encouraged that the reviewer finds our "contributions to be quite substantial", with a "good combination" of experiments and "relevant & meaningful" theories. We address Reviewer qXHf's concerns and questions below:
>
> ## Regarding paper format and presentations
>
> We apologize that trying to fit as many results as possible in the 10 pages did make the paper more dense and less easier to digest sometimes. Therefore, we are happy to take the reviewer's suggestion by omitting certain results in the main texts. Specifically, we plan to (1) shorten the results with longer horizons in Sec 3.2, which mainly serve as an ablation study and sanity check; (2) simplify the detailed introduction to our constructed counter-examples to fail advanced LLMs, where we believe conveying only the *intuition* for construction suffices; (3) move back the concluding remarks and remarks/discussions of the experiments,  so that the paper will not end abruptly. We hope these editions will make the paper less dense and more reader-friendly. If the reviewer also agrees with these suggestions, we will make the revision immediately.

---

> ### Author Response · Authors · 2024-11-21
> **Rebuttal by Authors (2)**
>
> ## Regarding the underlying assumption of Proposition 1
>
> Our *trend-checking* framework was meant to be designed for *general* sequences $\\{a_t\\}\_{t=1}^T$ for which we **do not know beforehand** how they were generated, since in the Online Learning setting, by definition, there should be *no* prior assumption on how $\\{\text{Regret}_t/t\\}\_{t=1}^T$ is generated, which very much depends on *both* how the loss sequences and how the policies are generated (by the algorithms). In order to leverage quantitative statistical methods, such as *Hypothesis Testing* or *Regression* (see our regression-based framework), to understand the experimental results statistically, one may have to make some underlying statistical assumptions.
>
>
> As the reviewer correctly pointed out, this approach implicitly assumes that $\\{(a_{t+1} - a_t)\\}\_{t=1}^T$ are independent. We used this assumption since without knowing how $\\{\text{Regret}_t/t\\}\_{t=1}^T$ were generated, one *possible* (statistical) assumption to *model arbitrarily changing sequences* is that at each $t$, some new element is generated randomly and independently, **without being *affected/biased*** by any previous elements in the sequence (since we do not know a priori how to model it). Indeed, even for our regression-based framework, which may be more intuitive to understand, we know that some statistical assumptions on the residuals are needed implicitly, see e.g., [4]. Hence, technically, these frameworks may not be applicable to *every possible* "loss-sequence" +  "algorithm/LLM-agent" combination (e.g., some non-stochastic and adversarial corner cases). That being said, these frameworks should still be useful as *indicators* for the no-regret behaviors, and this was also exactly why we developed **more than one** frameworks, and more importantly, **also compared with known no-regret algorithms FTRL/FTPL**, when evaluating the no-regret behaviors of LLMs. We have added such clarifications in the updated paper. We really appreciate your feedback on this point.
>
> Moreover, motivated by your feedback, we have calculated the correlation between
> $$
> \Delta_t = \frac{\text{Regret}\_t}{t} - \frac{\text{Regret}\_{t+1}}{t+1},
> $$
> by treating $(\Delta_t)_{t=1}^T$ as random variables. We have summarized the result in Appendix C.3.
>
> Let us revisit the proof of Proposition 1. We utilized the following equation:
> $$
> \mathbb{P}\left(\sum_{t=1}^T \pmb{1}[\Delta_t>0]> k\right) \underset{(i)}{=} \sum_{k = s}^{T-1} p^k (1-p)^{T-1-k} {T-1\choose k}
> $$
> while step $(i)$ holds under the assumption that $(\pmb{1}[\Delta_t>0])_{t=1}^T$ are independent. If $(\pmb{1}[\Delta_t>0])\_{t=1}^T$ are not independent, this equality might not hold. Nevertheless, since $\pmb{1}[\Delta_t>0]$ is a binary random variable, by the empirical validation of the weak correlation in Appendix C.3, we have
>
> $$
> \mathbb{E}\left[ \sum_{t=1}^T \pmb{1}[\Delta_t>0] \right] = \sum_{t=1}^{T}\mathbb{E} \left[\pmb{1}[\Delta_t>0]\right],
> $$
> $$
> \text{Var}\left(\sum_{t=1}^T \pmb{1}[\Delta_t>0]\right)\approx \sum_{t=1}^{T}\text{Var} \left(\pmb{1}[\Delta_t>0]\right).
> $$
>
> This implies that the random variable $\sum_{t=1}^T \pmb{1}[\Delta_t>0]$ indeed has the same first order and second order moment as the case that those random variable $(\pmb{1}[\Delta_t>0])$ ($t \in [1,T]$) are independent. Therefore, we regard a Binomial distribution (i.e., assuming $\{\pmb{1}[\Delta_t>0]\}$  ($t \in [1,T]$) to be independent) to be an acceptable approximation for the actual random variable $\sum_{t=1}^T \pmb{1}[\Delta_t>0]$, which finally gives step $(i)$. In fact, when binary random variables have weak correlations (but are not necessarily independent), using the Binomial distribution as an approximation for their sum is common in the engineering literature [5]. We sincerely appreciate this comment that has helped improve our paper.

---

> ### Author Response · Authors · 2024-11-21
> **Rebuttal by Authors (3)**
>
> ## Regarding LLMs at different stages
>
> We indeed agree with the reviewer that in practice, training an LLM often involves two stages, pre-training and fine-tuning, where fine-tuning can further incorporate several methods, like instruction-tuning, RLHF, etc. Meanwhile, the actual training pipeline can also be significantly different among different LLMs.
>
> Therefore, when trying to understand the trained foundation models for decision making, theoretically, it is standard (see e.g., [1, 2, 3]) to analyze the maximum likelihood objective (Eq. 4.1) as in our Sec 4.2, which can not only model the pre-training stage, but also certain fine-tuning stages, like instruction-tuning (i.e., predict the label $y$ given input question $x$.). In other words, such **supervised fine-tuning** stages like instruction-tuning also fit in our framework in Sec 4.2 as long as suitable labels/supervisions are provided in the instruction-tuning dataset.
>
> Finally, we do admit that there also exist more advanced fine-tuning methods like RLHF, whose objective is beyond a log likelihood maximization and next-token-prediction. We focused on such losses as an initial attempt towards understanding the no-regret behaviors of LLMs. We believe understanding the effects of these more advanced fine-tuning methods on the decision-making ability/no-regret properties of LLMs is an important open question.
>
> Again, we  really appreciate the feedback from Reviewer qXHf that has helped improve the paper. Please feel free to let us know if there are any other  questions.

---

> ### Author Response · Authors · 2024-11-21
> **Rebuttal by Authors (4)**
>
> ---
>
> [1] Jonathan Lee, et al. "Supervised pretraining can learn in-context reinforcement learning." Advances in Neural Information Processing Systems 36 (2024).
>
> [2] Zhihan Liu, et al. "Reason for future, act for now: A principled architecture for autonomous llm agents." Forty-first International Conference on Machine Learning. 2023.
>
> [3] Licong Lin, Yu Bai, and Song Mei. "Transformers as Decision Makers: Provable In-Context Reinforcement Learning via Supervised Pretraining." The Twelfth International Conference on Learning Representations.
>
> [4] https://en.wikipedia.org/wiki/Regression_analysis#Underlying_assumptions
>
> [5] Arnljot Hoyland and Marvin Rausand. "System reliability theory: models and statistical methods". John Wiley & Sons, 2009.

---

> ### Author Response · Authors · 2024-11-24
> **Comment**
>
> We sincerely appreciate the valuable feedback you provided. We wanted to follow up to ensure that our revisions have adequately addressed your concerns. If you have any additional comments or questions, we would be grateful if you could share them with us.
>
> Thanks for your time and considerations.
>
> Best regards,
>
> The Authors

---

> ### Author Response · Authors · 2024-12-03
> **Thanks for your feedback**
>
> Dear Reviewer qXHf,
>
> Thank you for your thoughtful feedback, especially on Proposition 1 and LLM training. We hope our follow-up comments have effectively addressed your concerns. As the discussion phase draws to a close, we wanted to check if you have any additional feedback or concerns. Please feel free to share, and we would be more than happy to discuss further. Thank you once again!
>
> Best regards,
>
> The authors

---

### Author Response · Authors · 2024-11-21
**General Response**

We appreciate all the feedback from the reviewers, and have addressed them carefully in each individual rebuttal. We here summarize several significant changes we made in the revised paper.

## Real-World Case Studies:

With the focus on **understanding** the online decision-making capabilities and strategic behaviors of LLMs, our extensive experiments were mostly on synthetic examples (as in other recent works with the same focus), in order to **have controlled experiments, establish some principles, and gain some theoretical insights**. That being said, we appreciate the comments from the reviewers, and have managed to include real-world case studies, on **Sequential Recommendations** and **Interactive Negotiation**, with real-world data. See Section C.11 for more details.

## Clarifications:

There was some confusion regarding the assumptions for our theoretical and validation framework, experimental setup, and use of some terminologies, for which we have added clarifications and made changes throughout.


Again, we sincerely appreciate the valuable feedback from the reviewers that has helped strengthen our paper. We would be more than happy to answer any further questions that the reviewers may have.

The Authors

---

### Meta-Review · Area_Chair_PeWA · 2024-12-19

**Metareview:**

Summary:
This work explores the use of large language model (LLM) agents in playing games. The authors conduct experiments across several LLM instances and problem settings, delivering key insights into LLM performance in gaming scenarios, such as demonstrating that most LLM agents can achieve no-regret performance. Furthermore, the authors propose a distributional assumption, the quantal response model, to characterize LLM response distributions and validate its effectiveness through experiments.

Strengths:

- The paper provides a theoretical framework to understand no-regret behaviors in LLMs and introduces a novel training objective, the regret-loss, with provable guarantees.
- The study features a diverse range of experiments, including synthetic and toy-case settings (arbitrary, non-stationary, and bandit feedback environments), and compares LLMs with well-established no-regret algorithms.

Weaknesses:

The presentation can be improved. Notably, the manuscript is approximately 80 pages long, which is unusually lengthy for a conference submission and may hinder accessibility.

Recommendation:
I recommend acceptance based on the strengths mentioned above. Despite the need for a more concise presentation, the paper offers valuable theoretical insights and empirical findings that contribute significantly to the understanding of LLMs in gaming scenarios.

**Additional Comments On Reviewer Discussion:**

During the discussion phase, the authors focused on improving the presentation of the paper, which I believe has significantly enhanced its readability and accessibility.

---

### Decision · Program_Chairs · 2025-01-22

Accept (Poster)